# A Markovian decision process for variable selection in Branch & bound

## Abstract

Mixed-Integer Linear Programming (MILP) is a powerful framework used to address a wide range of NP-hard combinatorial optimization problems, often solved by Branch and bound (B&B). A key factor influencing the performance of B&B solvers is the variable selection heuristic governing branching decisions. Recent contributions have sought to adapt reinforcement learning (RL) algorithms to the B&B setting to learn optimal branching policies, through Markov Decision Processes (MDP) inspired formulations, and ad hoc convergence theorems and algorithms. In this work, we introduce B&B MDPs, a principled vanilla MDP formulation for variable selection in B&B, allowing to leverage a broad range of RL algorithms for the purpose of learning optimal B&B heuristics. Computational experiments validate our model empirically, as our branching agent outperforms prior state-of-the-art RL agents on four standard MILP benchmarks.

## 1 Introduction

Mixed-Integer Linear Programming (MILP) is a subfield of combinatorial optimization (CO), a discipline that aims at finding solutions to optimization problems with large but finite sets of feasible solutions. Specifically, mixed-integer linear programming addresses CO problems that are NP-hard, meaning that no polynomial-time resolution algorithm has yet been discovered to solve them. Mixed-integer linear programs are used to solve efficiently a vast range of high-dimensional combinatorial problems, spanning from operations research (Hillier & Lieberman, 2015) to the fields of deep learning (Tjeng et al., 2019), finance (Mansini et al., 2015), computational biology (Gusfield, 2019), and fundamental physics (Barahona, 1982). MILPs are traditionally solved using Branch and bound (B&B) (Land & Doig, 1960), an algorithm which methodically explores the space of solutions by dividing the original problem into smaller sub-problems, while ensuring the optimality of the final returned solution. Intensively developed since the 1980s (Bixby, 2012), MILP solvers based on the B&B algorithm are high-performing tools. In particular, they rely on complex heuristics fine-tuned by experts on large heterogeneous benchmarks (Gleixner et al., 2021). Hence, in the context of real-world applications, in which similar instances with slightly varying inputs are solved on a regular basis, there is a huge incentive to reduce B&B total solving time by learning efficient tailor-made heuristics. The branching heuristic, or variable selection heuristic, which determines how to iteratively partition the space of solutions, has been found to be critical to B&B computational performance (Achterberg & Wunderling, 2013). Over the last decade, many contributions have sought to harness the predictive power of machine learning (ML) to learn better-performing B&B heuristics (Bengio et al., 2021; Scavuzzo et al., 2024). By using imitation learning (IL) to replicate the behaviour of a greedy branching expert at lower computational cost, Gasse et al. (2019) established a landmark result as they first managed to outperform a solver relying on human-expert heuristics. Building on the works of Gasse et al. (2019) and He et al. (2014), who proposed a Markov decision process (MDP) formulation for node selection in B&B, several contributions succeeded in learning efficient branching strategies by reinforcement (Etheve et al., 2020; Scavuzzo et al., 2022; Parsonson et al., 2022), without surpassing the performance achieved by the IL approach. Yet,

if the performance of IL heuristics are caped by that of the suboptimal branching experts they learn from, the performance of RL branching strategies are, in theory, only bounded by the maximum score achievable. We note that in order to cope with dire credit assignment problems (Pignatelli et al., 2023) induced by the sparse reward model described in He et al. (2014), prior research has shifted away from the traditional Markov decision process framework, finding it impractical to learn efficient branching strategies. Instead, Etheve et al. (2020), Scavuzzo et al. (2022) and Parsonson et al. (2022) have adopted unconventional MDP inspired formulations to model variable selection in B&B.

In this work, we show that despite improving the convergence properties of RL algorithms, these alternative formulations introduce approximations which undermine the asymptotic performance of RL branching agents in the general case. In order to address this issue, we introduce branch and bound Markov decision processes (BBMDP), a principled vanilla MDP formulation for variable selection in B&B, which preserves convergence properties brought by previous contributions without sacrificing optimality. Our new formulation allows to define a proper Bellman optimality operator, which in turns enables to unlock the full potential of state-of-the-art approximate dynamic programming algorithms (Hessel et al., 2017; Dabney et al., 2018; Farebrother et al., 2024) for the purpose of learning optimal B&B branching strategies. We evaluate our method on four classic MILP benchmarks, achieving state-of-the art performance and dominating previous RL agents while narrowing the gap with the IL approach of Gasse et al. (2019), as shown in Figure 1.

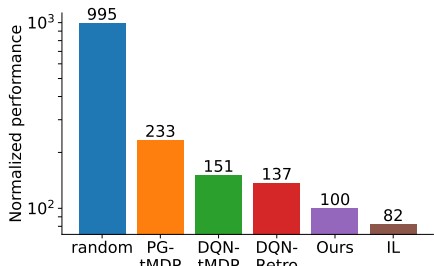

Figure 1: Normalized scores in log scale of IL, RL and random agents across the Ecole benchmark Prouvost et al. (2020).

## 2 PROBLEM STATEMENT

### 2.1 MIXED-INTEGER LINEAR PROGRAMMING

We consider mixed-integer linear programs (MILPs), defined as:

$$P : \begin{cases} \min c^\top x \\ l \leq x \leq u \\ Ax \leq b \; ; \; x \in \mathbb{Z}^{|\mathcal{I}|} \times \mathbb{R}^{n-|\mathcal{I}|} \end{cases}$$

with $n$ the number of variables, $m$ the number of linear constraints, $l, u \in \mathbb{R}^n$ the lower and upper bound vectors, $A \in \mathbb{R}^{m \times n}$ the constraint matrix, $b \in \mathbb{R}^m$ the right-hand side vector, $c \in \mathbb{R}^n$ the objective function, and $\mathcal{I}$ the indices of integer variables. Throughout this document, we are interested in repeated MILPs of fixed dimension $\{P_i = (A_i, b_i, c_i, l_i, u_i)\}_{i \in N}$ which are understood as realizations of a random variable following an unknown distribution $p_0 : \Omega \to \mathbb{R}^{m \times n} \times \mathbb{R}^m \times \mathbb{R}^n \times \mathbb{R}^n \times \mathbb{R}^n$.

In order to solve MILPs efficiently, the B&B algorithm iteratively builds a binary tree $(\mathcal{V}, \mathcal{E})$ where each node corresponds to a MILP, starting from the root node $v_0 \in \mathcal{V}$ representing the original problem $P_0$. The incumbent solution $\bar{x} \in \mathbb{Z}^{|\mathcal{I}|} \times \mathbb{R}^{n-|\mathcal{I}|}$ denotes the best feasible solution found at current iteration, its associated value $GUB = c^\top \bar{x}$ is called the *global upper bound* on the optimal value. The overall state of the optimization process is thus captured by the triplet $s = (\mathcal{V}, \mathcal{E}, \bar{x})$, we note $\mathcal{S}$ the set of all such triplets.[1] Throughout the optimization process, B&B nodes are explored sequentially. We note $\mathcal{C}$ the set of visited or closed nodes, and $\mathcal{O}$ the set of unvisited or open nodes, such that $\mathcal{V} = \mathcal{C} \cup \mathcal{O}$. Originally, $\mathcal{O} = \{v_0\}$ and $\mathcal{C} = \emptyset$. At each iteration, the node selection policy $\rho : \mathcal{S} \to \mathcal{O}$ selects the next node to explore. Since $\rho$

---

[1]To account for early resolution steps where no incumbent solution has yet been found, we define a special value for $\bar{x}$, whose $GUB = \infty$. For the sake of simplicity, we make this implicit in the remainder of the paper.

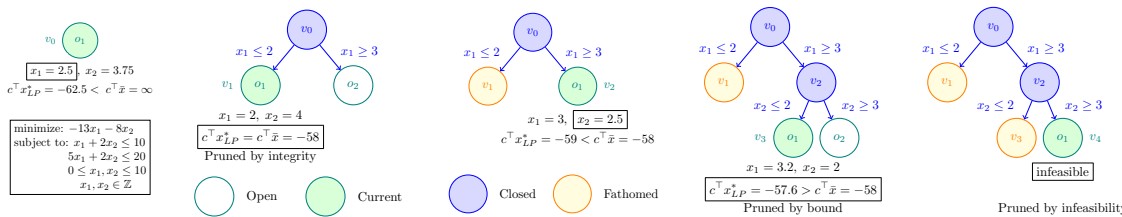

Figure 2: Solving a MILP by B&B using variable selection policy $\pi$ and node selection policy $\rho$. Each node $v_i$ represents a MILP derived from the original problem, each edge represents the bound adjustment applied to derive child nodes from their parent. At each step, nodes $o_i \in \mathcal{O}$ are re-indexed according to $\rho$.

necessarily defines a total order on nodes $o_i \in \mathcal{O}$, we can arrange indices such that $o_1 = \rho(s)$ denotes $v_t$ the node currently explored at step $t$. Figure 2 illustrates how B&B operates on an example. At each iteration, let $x_{LP}^* \in \mathbb{R}^n$ be the optimal solution to the linear relaxation of $P_t$, the problem associated with $v_t$:

- If $P_t$ admits no solution, $v_t$ is marked as visited and the branch is pruned by infeasibility. If $x_{LP}^* \in \mathbb{R}^n$ exists, and $GUB < c^\top x_{LP}^*$, no integer solution in the subsequent branch can improve $GUB$, thus $v_t$ is marked as closed and the branch is pruned by bound. If $x_{LP}^*$ is not dominated by $\bar{x}$ and $x_{LP}^*$ is feasible (all integer variables in $x_{LP}^* \in \mathbb{R}^n$ have integer values), a new incumbent solution $\bar{x} = x_{LP}^*$ has been found. Hence $GUB$ is updated and $v_t$ is marked as visited while the branch is pruned by integrity.

- Else, $x_{LP}^*$ admits fractional values for some integer variables. The branching heuristic $\pi : \mathcal{S} \to \mathcal{I}$ selects a variable $x_b$ with fractional value $\hat{x}_b$, to partition the solution space. As a result, two child nodes $(v_-, v_+)$, with associated MILPs $P_- = P_t \cup \{x_b \leq \lfloor \hat{x}_b \rfloor\}$ and $P_+ = P_t \cup \{x_b \geq \lceil \hat{x}_b \rceil\}$, are added to the current node.[2] Their linear relaxation is solved, before they are added to the set of open nodes $\mathcal{O}$ and $v_t$ is marked as visited.

This process is repeated until $\mathcal{O} = \emptyset$ and $\bar{x}$ is returned. The dynamics of the B&B algorithm between two branching decisions can be described by the function $\kappa_\rho : \mathcal{S} \times \mathcal{I} \to \mathcal{S}$, such that $s' = \kappa_\rho(s, \pi(s))$. By design, B&B does not terminate before finding an optimal solution and proving its optimality. Consequently, optimizing the performance of B&B on a distribution of MILP instances is equivalent to minimizing the expected solving time of the algorithm. As Etheve (2021) evidenced, the variable selection strategy $\pi$ is by far the most critical B&B heuristic in terms of computational performance. In practice, the total number of nodes of the B&B tree is used as an alternative metric to evaluate the performance of branching heuristics $\pi$, as it is a hardware-independent proxy for computational efficiency. Under these circumstances, given a fixed node selection strategy $\rho$, the optimal branching strategy $\pi^*$ associated with a distribution $p_0$ of MILP instances can be defined as:

$$\pi^* = \arg\min_\pi \mathbb{E}_{P \sim p_0}(|BB_{(\pi,\rho)}(P)|) \tag{1}$$

with $|BB_{(\pi,\rho)}(P)|$ the size of the $B\&B$ tree after solving $P$ to optimality following strategies $(\pi, \rho)$.

## 2.2 REINFORCEMENT LEARNING

We consider the setting of discrete-time, deterministic MDPs (Puterman, 2014) defined by the tuple $(\mathcal{S}, \mathcal{A}, \mathcal{T}, p_0, \mathcal{R})$. At each time step $t$, the agent observes $s_t \in \mathcal{S}$ the current state of the environment, before executing action $a_t \in \mathcal{A}$, and receiving reward $r_t = \mathcal{R}(s_t, a_t)$. The Markov transition function

---

[2]$\hat{x}_b$ denotes the value of $x_b$ in $x_{LP}^*$. We use the symbol $\cup$ to denote the refinement of the bound on $x_b$ in $P_t$.

$\mathcal{T} : \mathcal{S} \times \mathcal{A} \to \mathcal{S}$ models the dynamics of the environment. In particular, it satisfies the Markov property: conditionally to $s_t$ and $a_t$, $s_{t+1}$ is independent of all past visited states and actions. Given a trajectory starting in state $s_0$ sampled according to the initial distribution $p_0$, the total gain is defined for all $t \geq 0$ as $G_t = \sum_{t'=t}^{\infty} \gamma^{t'-t} \cdot \mathcal{R}(s_{t'}, a_{t'})$, with $\gamma \in [0, 1]$. The objective of an RL agent is to maximize the expected gain of the trajectories yielded by its action selection policy $\pi : \mathcal{S} \to \mathcal{A}$. This is equivalent to finding the policy maximizing value functions $V^\pi(s_t) = \mathbb{E}_{a_{t'} \sim \pi(s_{t'})}[G_t]$ and $Q^\pi(s_t, a_t) = \mathcal{R}(s_t, a_t) + \gamma \cdot V^\pi(s_{t+1})$. The optimal $Q$-value function $Q^*$ indicates the highest achievable cumulated gain in the MDP. It satisfies the Bellman optimality equation $Q(s, a) = \mathcal{R}(s, a) + \gamma \cdot \max_{a' \in \mathcal{A}} Q(\mathcal{T}(s, a), a')$, for $(s, a) \in \mathcal{S} \times \mathcal{A}$. The optimal policy is retrieved by acting greedily according to the learned $Q$-value function: $\pi^*(s) = \arg \max_{a \in \mathcal{A}} Q^*(s, a)$.

## 2.3 RELATED WORK

Following the seminal work by Gasse et al. (2019), few contributions have proposed to build more complex neural network architectures based on transformers (Lin et al., 2022) and recurrence mechanisms (Seyfi et al., 2023) to improve the performance of IL branching agents, with moderate success. In parallel, theoretical and computational analysis (Bestuzheva et al., 2021; Sun et al., 2022) have shown that neural networks trained by imitation could not rival the tree size performance achieved by strong branching (SB), the branching expert used in Gasse et al. (2019). In fact, low tree sizes associated with SB turn out to be primarily due to the formulation improvements resulting from the massive number of LPs solved in SB, not to the intrinsic quality of the branching decisions themselves.

Since branching decisions are made sequentially, reinforcement learning appears as a natural candidate to learn good branching policies. Etheve et al. (2020) and Scavuzzo et al. (2022) proposed the model of TreeMDP, in which state $s_i = (P_i, x^*_{LP,i}, \bar{x}_i)$ consists in the MILP associated with node $v_i$ along with the solution of its linear relaxation and the incumbent solution at $v_i$. The actions available at $s_i$ is the set of fractional variables in $x^*_{LP,i}$. Given $(s_i, a_i)$ the tree Markov transition function produces two child node states $(s_i^-, s_i^+)$ that can be visited in any order. Crucially, when the B&B tree is explored in depth-first-search (DFS), TreeMDP trajectories can be divided in independent subtrees, allowing to learn policies minimizing the size of each subtree independently. This helps mitigate credit assignment issues that arise owing to the length of episode trajectories. Subsequently, Parsonson et al. (2022) found the DFS node selection policy to be highly detrimental to the computational performance of RL branching strategies. Assuming that RL branching agents trained following advanced node selection strategies would perform better despite the lack of theoretical guarantee, they proposed to learn from retrospective trajectories, diving trajectories built from original TreeMDP episodes. In fact, Parsonson et al. (2022) found retrospective trajectories to alleviate the partial observability induced by the "disordered" exploration of the tree and outperform prior RL agents.

A large body of work has proposed to learn, either by imitation or reinforcement, better-performing B&B heuristics outside of variable selection (Nair et al., 2021; Paulus et al., 2022). RL contributions in primal search (Sonnerat et al., 2022; Wu & Lisser, 2023) node selection (He et al., 2014; Etheve, 2021) and cut selection (Tang et al., 2020; Song et al., 2020; Wang et al., 2023) have all relied on the TreeMDP framework to train their agents, simply adapting the action set to the task at hand. Finally, machine learning applications in combinatorial optimization are not limited to B&B. For example, in the context of routing or scheduling problems where exact resolution rapidly becomes prohibitive, agents are trained to learn direct search heuristics (Kool et al., 2019; Grinsztajn et al., 2022; Chalumeau et al., 2023) yielding high-quality feasible solutions.

## 3 BRANCH AND BOUND MARKOV DECISION PROCESS

By using the current B&B node as the observable state, prior attempts to learn optimal branching strategies have relied on the TreeMDP formalism to train RL agents. However, TreeMDPs are not MDPs, as they

do not define a Markov process on the state random variable (for instance, a transition yields two states and is hence not a stochastic process on the state variables). As a result, this forces Etheve et al. (2020) and Scavuzzo et al. (2022) to redefine Bellman updates and derive *ad hoc* convergence theorems for TD(0), value iteration, and policy gradient algorithms. In order to leverage broader theoretical results from the reinforcement learning literature, we propose a description of variable selection in B&B as a proper Markov decision process.

### 3.1 DEFINITION

The problem of finding an optimal branching strategy according to Equation (1) can be described as a regular deterministic Markov decision process. To this end, we introduce Branch and bound Markov decision processes (BBMDP) by making the tuple $(\mathcal{S}, \mathcal{A}, \mathcal{T}, p_0, \mathcal{R})$ explicit, taking $\gamma = 1$ since episodes horizons are bounded by the (finite) largest possible number of nodes:

- **State space.** $\mathcal{S}$ is the set of all B&B trees $s_t = (\mathcal{V}_t, \mathcal{E}_t, \bar{x}_t)$. Note that this includes intermediate B&B trees, whose incumbent solutions $\bar{x}_t$ are yet to be proven optimal.

- **Action space.** $\mathcal{A}$ is the set of all integer variables indices $\mathcal{I}$.

- **Transition function**: The Markov transition function is defined as $\mathcal{T} = \kappa_\rho$ with $\kappa_\rho$ the branching operation described in Section 2.1. Note that if the variable associated with $a_t$ is not fractional in $x^*_{LP,t}$, then $s_{t+1} = \mathcal{T}(s_t, a_t) = s_t$ as relaxing a variable that is not fractional has no impact on the LP relaxation. Importantly, all states for which $\mathcal{O} = \emptyset$ are terminal states.

- **Starting states.** Initial states are single node trees, where the root node is associated to a MILP $P_0$ drawn according to the distribution $p_0$ defined in Section 2.1 (hence the use of $p_0$ for both the initial problem $P_0$ and the MDP's initial state $s_0$).

- **Reward model.** We define $\mathcal{R}(s, a) = -2$ for all transitions until episode termination. Since each transition results in the addition of two B&B nodes, the overall value of a trajectory is the opposite of the number of node added to the B&B tree from the root node, which is inline with the definition of Equation (1).

Unlike in TreeMDP, the current state is defined as the state of the entire B&B tree, rather than merely the current B&B node. The transition function returns a B&B tree whose open nodes are sorted according to the node selection policy $\rho$, thus reflecting the true dynamics of the B&B algorithm, instead of a couple of pseudo-states associated with former current node's child nodes. Note that the definition above sets BBMDPs among the specific class of MDPs called stochastic shortest path problems (Puterman, 2014).

### 3.2 LEARNING OPTIMAL BRANCHING STRATEGIES WITH BBMDPs

Like in TreeMDP, episode trajectories can be decomposed in independent subtree trajectories, to facilitate RL agents training. Let us consider $\pi$ a deterministic branching policy, we rewrite $V^\pi$ and $Q^\pi$ to exhibit their tree structure. Given a node $v \in \mathcal{V}_t$, we note $T(v)$ the subtree rooted in $v$. Noting $\mathcal{M} = \mathcal{O} \times \mathbb{R}^n$, we define $W^\pi : \mathcal{S} \times \mathcal{M} \to \mathbb{R}$ the $W$-value function that returns the opposite of the size of the subtree rooted in $o_i \in \mathcal{O}_t$ when branching according to policy $\pi$ starting from state $s_t$ until the episode termination. Importantly, $W^\pi$ depends on $\bar{x}_{o_i} \in \mathbb{R}^n$, the incumbent solution when $o_i$ is processed by the branch and bound algorithm. Then $V^\pi$ can be expressed as:

$$V^\pi(s_t) = Q^\pi(s_t, \pi(s_t)) = \sum_{o_i \in \mathcal{O}_t} W^\pi(s_t, o_i, \bar{x}_{o_i}) \tag{2}$$

To put it simply into words, the total number of nodes that will be added to the B&B tree past $s_t$ is equal to the sum of the sizes of all the subtrees $T(o_i)$ rooted in the open nodes of $s_t$. It is tempting to define $W$-value

functions merely as functions of $(o_i, \bar{x}_{o_i})$ for $o_i \in \mathcal{O}_t$ rather than functions of $(s_t, o_i, \bar{x}_{o_i})$, which comprises the whole B&B tree. The rationale for such value functions is that the size of the subtree rooted in $o_i \in \mathcal{O}_t$, for a given incumbent solution $\bar{x}_{o_i}$, should be the same, regardless of the parents of $o_i$, its position in the tree, or the branching decisions taken in subtrees $T(o_j)$ for $o_j \in \mathcal{O}_t$ and $j \neq i$. It turns out, this last statement does not always hold, quite counter-intuitively. Let us write $\tau_i$ the time steps at which the nodes $o_i \in \mathcal{O}_t$ are selected by the node selection strategy $\rho$.[3] Now, consider for instance a node selection procedure $\rho$ that performs a breadth-first search through the tree. The number of nodes in $T(o_i)$ will depend strongly on whether an improved incumbent solution $\bar{x}_{o_i}$ was found in the subtrees explored between $s_t$ and $s_{\tau_i}$, and in turn from the branching decisions taken in these subtrees. This example highlights the major issue of the node selection strategy $\rho$, when one wishes to define subtree sizes based on $(o_i, \bar{x}_{o_i})$.

Consider now two open nodes $o_i$ and $o_j$ in $\mathcal{O}_t$. Conversely to the previous example, if one can guarantee that the subtree rooted in $o_j$ will be solved to optimality before $o_i$ is considered for expansion in the B&B process, then the number of nodes in $T(o_i)$ will not be affected by the branching decisions taken at any node under $o_j$. In fact, if $o_j$ is solved to optimality, $\bar{x}_{o_i}$ will either not change if no feasible solution in $T(o_j)$ improves $GUB$, or either be the best feasible solution of the MILP associated with $o_j$, which does not depend on the series of actions taken in $T(o_j)$. In other words, to make sure that the size of $T(o_i)$ does only depend on the branching decisions taken in $T(o_i)$, all nodes $o_j \in \mathcal{O}_t$ must have been either fully explored or strictly unexplored at $\tau_i$. Applying this argument recursively induces that the only node selection strategy which enables predicting a subtree size only based on $(o_i, \bar{x}_{o_i})$, is a depth-first search (DFS) exploration of the B&B tree. The same observation was made by Etheve et al. (2020) and Scavuzzo et al. (2022) previously.

Therefore, we consider $\rho = DFS$ and write $W^\pi(M_t^i)$ the opposite of the size of $T(o_i)$ for $o_i \in \mathcal{O}_t$, with $M_t^i = (o_i, \bar{x}_{o_i}) \in \mathcal{M}$. We can now derive a refined Bellman update to train branching agents in BBMDP.

**Proposition 1.** *In BBMDP, the Bellman equation $V^\pi(s_t) = \mathcal{R}(s_t, a_t) + V^\pi(s_{t+1})$ writes:*

$$W^\pi(M_t^1) = -2 + W^\pi(M_{t+1}^1) + W^\pi(M_{t+1}^2) \tag{3}$$

*Proof.* (3) follows directly from injecting (2) in the Bellman equation, and observing that most terms in the sums simplify as $W^\pi(M_t^i) = W^\pi(M_{t+1}^{i+1})$ for $i \geq 2$. $\qquad\square$

In the following, we define $Q_\dagger^\pi : \mathcal{M} \times \mathcal{A} \to \mathbb{R}$ such that $Q_\dagger^\pi(M_t^1, a) = -2 + W^\pi(M_{t+1}^1) + W^\pi(M_{t+1}^2)$. Note that if $W^\pi$ and $Q_\dagger^\pi$ are not strictly value functions, they naturally appear when applying Bellman equations to BBMDP value functions under $\rho = DFS$. Importantly, we stress that in order to learn $\pi^*$, it is not necessary to learn $Q^*$, as you can deduce $\pi^*$ from $Q_\dagger^*$, which is both easier to manipulate as it only depends on quantities observable at $s_t$, and easier to learn as it trains on much shorter trajectories. In fact, $\pi^*(s) = \arg\max_{a \in \mathcal{A}} Q^*(s, a) = \arg\max_{a \in \mathcal{A}} Q_\dagger^*(M_s^1, a)$, with $M_s^1 = (o_1, \bar{x}_s)$ for $o_1 \in \mathcal{O}_s$ and $s \in \mathcal{S}$.

## 3.3 Q-LEARNING

We now propose to learn $\pi^*$ by training a neural network to approximate $Q_\dagger^*$ with traditional temporal difference (TD) algorithms. Consider a transition $(s, a, r, s') \in \mathcal{S} \times \mathcal{A} \times \mathbb{R} \times \mathcal{S}$. Applying the Bellman optimality operator $\mathcal{B}^*$, we obtain:

$$Q(s,a) = \mathcal{B}^*(Q(s, a)) \overset{3}{\iff} Q_\dagger(M_s^1, a) = -2 + \max_{a', a'' \in \mathcal{A}} Q_\dagger(M_{s'}^1, a') + Q_\dagger(M_{s'}^2, a'') \tag{4}$$

with $s' = \kappa_\rho(s, a)$. Our objective is to approximate, with a neural network $\mathbf{q}_\theta : \mathcal{M} \times \mathcal{A} \to \mathbb{R}$ parameterized by $\theta$, the value $Q_\dagger^{\pi_\theta}$ associated with policy $\pi_\theta$. Noting $s^{(k)}$ the state visited when following $\pi$ for $k-1$ steps

---

[3]Following our indexation of $o_i \in \mathcal{O}_t$, we have $t = \tau_1 < ... < \tau_i < ... < \tau_{|\mathcal{O}_t|}$.

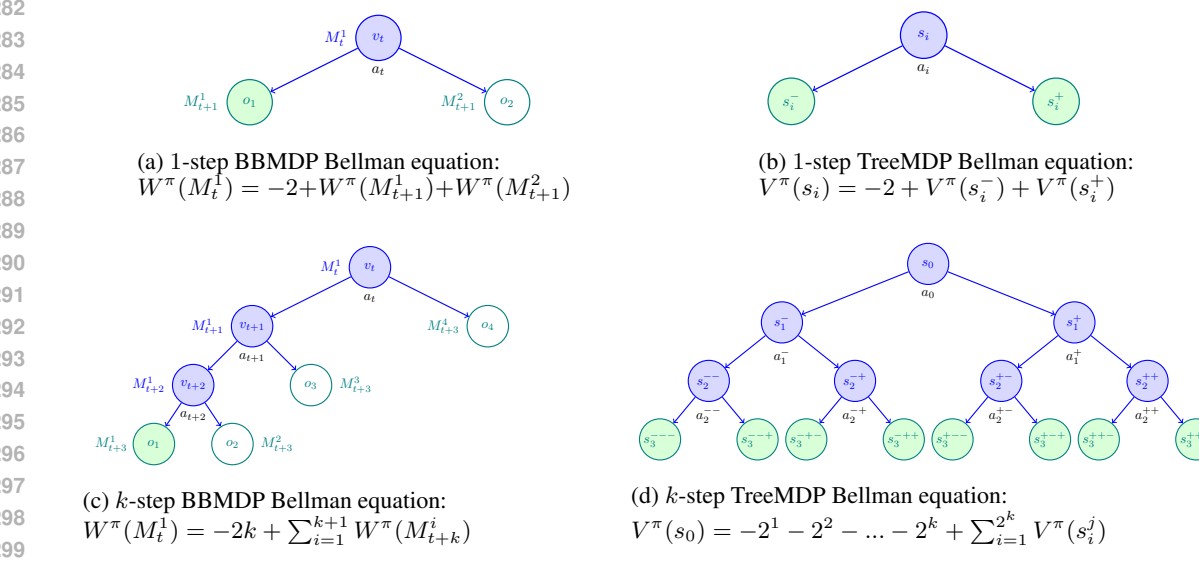

(a) 1-step BBMDP Bellman equation:
$W^\pi(M_t^1) = -2 + W^\pi(M_{t+1}^1) + W^\pi(M_{t+1}^2)$

(b) 1-step TreeMDP Bellman equation:
$V^\pi(s_i) = -2 + V^\pi(s_i^-) + V^\pi(s_i^+)$

(c) $k$-step BBMDP Bellman equation:
$W^\pi(M_t^1) = -2k + \sum_{i=1}^{k+1} W^\pi(M_{t+k}^i)$

(d) $k$-step TreeMDP Bellman equation:
$V^\pi(s_0) = -2^1 - 2^2 - ... - 2^k + \sum_{i=1}^{2^k} V^\pi(s_i^j)$

Figure 3: When applying TD(0), TreeMDP and BBMDP yield equivalent results, see 3a, 3b. However, when minimizing the $k$-step temporal difference loss, the two methods diverge as exemplified in 3c, 3d.

after performing action $a$ in $s$, we seek to minimize the $k$-step temporal difference loss:

$$\mathcal{L}_{MSE}(\mathbf{q}_\theta, Q_\dagger^{\pi_\theta}) = \mathbb{E}_{(s,a)\sim\pi_\theta}\left[\left(\mathbf{q}_\theta(M_s^1, a) - \left(-2k + \sum_{i=1}^{k+1} \max_{a'\in\mathcal{A}} Q_\dagger^{\pi_\theta}(M_{s(k)}^i, a')\right)\right)^2\right] \quad (5)$$

where we use $\mathbf{q}_\theta$ to bootstrap the values of $(Q_\dagger^{\pi_\theta}(M_{s(k)}^i, a))_{1\leq i\leq k+1}$. Following the work of Farebrother et al. (2024) on training value functions via classification, we introduce a HL-Gauss cross-entropy loss adapted to the B&B setting:

$$\mathcal{L}_{CE}(\mathbf{q}_\theta, Q_\dagger^{\pi_\theta}) = \mathbb{E}_{(s,a)\sim\pi_\theta}\left[\mathbf{q}_\theta(M_s^1, a) \cdot \log p_{hist}\left[\left(-2k + \sum_{i=1}^{k+1} \max_{a'\in\mathcal{A}} Q_\dagger^{\pi_\theta}(M_{s(k)}^i, a')\right)\right]\right] \quad (6)$$

where $p_{hist}$ is the function encoding $Q$-values into histogram categorical distributions, see Appendix E for complete description as well as theoretical motivation.

### 3.4 BBMDP VS TREEMDP

As it evacuates the core MDP notions of temporality and sequentiality, TreeMDP fails to describe variable selection in B&B accurately in the general case. This is illustrated in Figure 3: although the TreeMDP model is a valid approximation of BBMDP when training an RL agent to minimize the one-step temporal difference, it produces inconsistent learning schemes when considering a multi-step temporal difference loss. In fact, applying Etheve (2021) tree Bellman operator repeatedly yields trees that cannot be produced by a B&B algorithm explored in DFS.

Hence, BBMDP leverages the results established in Etheve et al. (2020) and Scavuzzo et al. (2022)–in DFS, acting according to a policy minimizing the size of the subtree rooted in the current B&B node is

equivalent to acting according to a global optimal policy–all while preserving MDP properties. Crucially, BBMDP allows to harness RL algorithms that are not compatible with the TreeMDP framework, such as $k$-step temporal difference, TD($\lambda$), or any RL algorithms using MCTS as policy improvement operator (Grill et al., 2020). In the same fashion, BBMDP can be applied to augment the pool of RL algorithms available for learning improved cut selection and primal search heuristics, simply by adapting the action set and the reward model to the task at hand.

## 4 EXPERIMENTS

We now compare our branching agent against prior IL and RL approaches. For our experiments, we use the open-source solver SCIP 8.0.3 (Bestuzheva et al., 2021) as backend MILP solver, along with the Ecole library (Prouvost et al., 2020) both for instance generation and environment simulation. We will make our code available upon publication.

### 4.1 EXPERIMENTAL SETUP

**Benchmarks** We consider four standard MILP benchmarks for learning branching strategies: set covering, combinatorial auctions, maximum independent set and multiple knapsack problems. We train and test on instances of same dimensions as Gasse et al. (2019), see Appendix A. As to SCIP configuration, we set the time limit to one hour, disable restart, and deactivate cut generation beyond root node. All the other parameters are left at their default value.

**Baselines** We compare our DQN-BBMDP agent against DQN-TreeMDP (DQN-tMDP) (Etheve et al., 2020) and REINFORCE-TreeMDP (PG-tMDP) (Scavuzzo et al., 2022) agents. We also compare against the IL expert from Gasse et al. (2019), and against DQN-Retro (Parsonson et al., 2022) the current state-of-the-art RL branching agent. More details on these baselines can be found in Appendix D. Finally, we report the performance of reliability pseudo cost branching (RPB), the default branching heuristic used in SCIP, strong branching (Applegate et al., 1995), the greedy expert from which the IL agent learns from, and random branching, which randomly selects a fractional variable.

### 4.2 TRAINING

**Network architecture** Following prior works, we use the bipartite graph representation introduced by Gasse et al. (2019) augmented by the features proposed in Parsonson et al. (2022) to represent B&B nodes. Additionally, we use the Gasse et al. (2019) graph convolutional architecture to parameterize our $Q$-value network, see Appendix C for a more detailed description.

**Learning algorithm** We train our $Q$-learning agent following a lightened version of Rainbow-DQN (Hessel et al., 2017), see Appendix B for a comprehensive description. Contrary to DQN-tMDP and DQN-Retro, we train our agent using the HL-Gauss cross-entropy loss described in section 3.3.

**Training & evaluation** Models are trained on easy instances of each benchmark separately, and evaluated on easy, medium and hard instances. Validation curves can be found in Appendix G. For evaluation, we report the node and time performance over 100 easy test instances unseen during training, as well as on 100 medium and 100 hard transfer instances of higher dimensions, see Table 3 in Appendix A. At evaluation, performance scores are averaged over 5 seeds. Importantly, when comparing a machine learning (IL or RL) branching strategy with a standard SCIP heuristic such as RPB or SB, time performance is the only relevant criterion. In fact, when implementing one of its own branching rules, SCIP triggers a series of techniques strengthening the current MILP formulation. If these techniques effectively reduce the number

| Method | Set Covering | | Comb. Auction | | Max. Ind. Set | | Mult. Knapsack | | Norm. Score | |
|---|---|---|---|---|---|---|---|---|---|---|
| | Node | Time | Node | Time | Node | Time | Node | Time | Node | Time |
| Presolve | — | 4.74 | — | 0.90 | — | 1.78 | — | 0.20 | — | — |
| Random | 3289 | 5.94 | 1111 | 2.16 | 386.8 | 2.01 | 733.5 | 0.55 | 995 | 374 |
| SB | 35.8 | 12.93 | 28.2 | 6.21 | 24.9 | 45.87 | 161.7 | 0.69 | 36 | 2358 |
| RPB | 62.0 | 2.27 | 20.2 | 1.77 | 19.5 | 2.44 | 289.5 | **0.53** | 51 | 253 |
| IL | **133.8** | 0.90 | **83.6** | 0.73 | **40.1** | **0.44** | 272.0 | 1.02 | 82 | 113 |
| IL-DFS | 136.4 | **0.74** | 95.5 | 0.67 | 69.4 | 0.56 | 472,8 | 1.54 | 114 | 129 |
| PG-tMDP | 649.4 | 2.32 | 168.0 | 0.94 | 153.6 | 0.92 | 436.9 | 1.57 | 233 | 206 |
| DQN-tMDP | 175.8 | 0.83 | 203.3 | 1.11 | 168.0 | 1.00 | 266.4 | 0.73 | 151 | 136 |
| DQN-Retro | 183.0 | 1.14 | 103.2 | 0.78 | 223.0 | 1.81 | 250.3 | 0.67 | 137 | 160 |
| DQN-BBMDP | **152.3** | **0.77** | **97.9** | **0.62** | **103.2** | **0.69** | **236.6** | **0.66** | **100** | **100** |

Easy instances (Test)

| Method | Set Covering | | Comb. Auction | | Max. Ind. Set | | Mult. Knapsack | | Norm. Score | |
|---|---|---|---|---|---|---|---|---|---|---|
| | Node | Time | Node | Time | Node | Time | Node | Time | Node | Time |
| Presolve | - | 12.3 | - | 2.67 | - | 5.16 | - | 0.46 | — | — |
| Random | 271632 | 842 | 317235 | 749 | 215879 | 2102 | 93452 | 70.6 | 5555 | 2737 |
| SB | 672.1 | 398 | 389.6 | 255 | 169.9 | 2172 | **1709** | **12.5** | 9 | 1425 |
| RPB | 3309 | 48.4 | 1376 | 14.77 | 3368 | 90.0 | 30620 | 22.1 | 62 | 90 |
| IL | **2610** | 23.1 | **1309** | **9.8** | **1882.0** | **37.6** | 9747 | 46.5 | **39** | **55** |
| IL-DFS | 3103 | **22.5** | 1802 | 11.1 | 3501 | 55.5 | 43224 | 177 | 75 | 93 |
| PG-tMDP | 44649 | 221 | 6001 | 30.7 | **3133** | **39.5** | 35614 | 165 | 298 | 233 |
| DQN-tMDP | 8632 | 71.3 | 20553 | 116 | 45634 | 477 | **22631** | **65.1** | 439 | 445 |
| DQN-Retro | 6100 | 59.4 | 2908 | 18.4 | 119478 | 1863 | 27077 | 79.5 | 494 | 662 |
| DQN-BBMDP | **5651** | **46.4** | **2273** | **11.8** | 7168 | 81.3 | 37098 | 109 | **100** | **100** |

Medium instances (Transfer)

Table 1: Performance comparison of branching agents on four standard MILP benchmarks. For each method, we report total number of B&B nodes, presolve time and total solving time outside of presolve. Lower is better, **red** indicates best agent overall, **blue** indicates best among RL agents. Presolve is common to all methods. Following prior works, we report geometrical mean over 100 easy instances unseen during training and over 100 higher-dimensional medium instances. Norm. Score denotes the aggregate average performance obtained by each agent across the four MILP benchmarks, normalized by the score of DQN-BBMDP.

of nodes to visit, they incur computational overhead which ultimately increases SCIP overall solving time. This renders node comparisons between ML and non-ML branching strategies negligible relative to solving time evaluations, as observed by Gamrath & Schubert (2018); Scavuzzo et al. (2022).

## 4.3 RESULTS

Computational results obtained on the four benchmarks are presented in Table 1. Additional performance metrics as well as further computational results on higher-dimensional instances are provided in Appendix H. On easy instances, DQN-BBMDP consistently obtains best performance among RL agents, while also outperforming IL-DFS agent. When compared against prior state-of-the-art DQN-Retro, DQN-BBMDP achieves an aggregate average 27% reduction of total number of node and 38% reduction of solving time outside presolve across the four Ecole benchmarks, see Figure 1. Contrary to Parsonson et al. (2022), we

|          | DQN-BBMDP | DQN-TreeMDP | DQN-BBMDP w.o. HL-Gauss | DQN-BBMDP w.o. DFS |
|----------|-----------|-------------|-------------------------|--------------------|
| $k = 1$  | 158.9     | 175.8 $(+10\%)$ | 169, 4 $(+7\%)$     | 156.2 $(-2\%)$     |
| $k = 3$  | **152.3** $(-4\%)$ | 178.9 $(+13\%)$ | 172.3 $(+8\%)$  | 150.1 $(-5\%)$     |

Table 2: Ablation impact of BBMDP, HL-Gauss loss and DFS. We remove one component one at the time, and evaluate corresponding versions on 100 easy set covering instances after training for $200,000$ gradient steps as described in 4.2.

find DQN-Retro to yield performance comparable to DQN-tMDP. Remarkably, all RL agents outperform the SCIP solver on 3 out of 4 benchmarks in terms of solving time. Although Gasse et al. (2019) IL agent remains the most efficient branching agent, the node gap between RL and IL across all four benchmarks has been more than halved, as shown in Figure 1.

On medium instances, DQN-BBMDP also dominates among RL agents, although it is outperformed by PG-tMDP on maximum independent set instances and by DQN-Retro on multiple knapsack instances. The aggregate performance gap between DQN-BBMDP and other RL baselines is notably wider on medium instances, which aligns with the advantages of using a principled MDP formulation over TreeMDP. In fact, DQN-BBMDP is the first RL agent to demonstrate robust generalization capabilities on medium instances, outperforming SCIP on 3 out of 4 benchmarks.

## 4.4 ABLATION STUDY

We perform an ablation study on easy set covering instances to separate the performance gain associated with BBMDP and the HL-Gauss classification loss. Since BBMDP and TreeMDP yield strictly equivalent learning schemes when minimizing one-step temporal difference, see Figure 3, we evaluate the performance gap between one-step and $k$-step TD learning for both DQN-BBMDP and DQN-TreeMDP.

As shown in Table 2, we find that the bulk of the performance gain is brought by the use of a cross-entropy loss. Nonetheless, we find that the use of a multi-step TD loss improves the performance of DQN-BBMDP, while it undermines the performance of DQN-TreeMDP. This further supports the adoption of BBMDP for learning branching strategies by reinforcement in the future. Following Parsonson et al. (2022), we also evaluate the cost of opting for depth-first search instead of best estimate search, SCIP's default node selection policy, when learning branching strategies. Contrary to their work, we find DFS not to be restrictive in practice in terms of performance. We further investigate theses discrepancies in Appendix F.

## 5 CONCLUSION AND PERSPECTIVES

Combinatorial optimization has proven to be a challenging setting for RL algorithms, including beyond the field of mixed-integer linear programming (Berto et al., 2023). Not only are RL agents rather consistently outperformed by human-expert CO heuristics or IL agents trained to mimic these experts, but their application has also been limited so far to fairly easy problem instances. In this work, we showed the theoretical and practical limits of the concept of TreeMDP for learning optimal branching strategies in MILP. Introducing BBMDP, we proposed a rigorous description of variable selection in B&B which we found to yield better performance than prior RL agents on the Ecole benchmark. We believe that building on a robust MDP formulation of variable selection in B&B is key to achieve substantial acceleration of solving time for higher-dimensional MILPs in the future. In fact, agents trained by reinforcement have proven in the past to be able to defeat human knowledge in combinatorial settings such as board games (Silver et al., 2017; Schrittwieser et al., 2020). Through our contribution, we built a theoretical framework that enables the adaptation of model-based MCTS RL algorithms for the purpose of learning optimal branching strategies.

## REPRODUCIBILITY STATEMENT

We provide detailed descriptions of our training algorithms, experimental setups, and network architectures in Section 4 and Appendix C. Furthermore, we provide all experiments details, including a full list of hyper-parameters, in Appendix B. We shared in the supplementary material an anonymized version of our code to reproduce the main experiments. We will share our open-source implementation to the community upon publication to facilitate future extensions.

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

APPENDIX

## A  INSTANCE DATASET

Instance datasets used for training and evaluation are decribed in Table 3. We trained and tested on instances of same dimensions as Gasse et al. (2019), Scavuzzo et al. (2022) and Parsonson et al. (2022). As a reminder, the size of action set $\mathcal{A}$ is equal to the number of integer variables in $P$. Consequently, action set sizes in the Ecole benchmark range from 30 to 480 for easy instances, from 50 to 980 for medium instances, and from 100 to 1480 for hard instances.

| Benchmark | Generation method | Parameters | Parameter value | | | # Int. variables | | |
|---|---|---|---|---|---|---|---|---|
| | | | Easy | Medium | Hard | Easy | Medium | Hard |
| Combinatorial auction | Leyton-Brown et al. (2000) | Items Bids | 100 500 | 200 1000 | 300 1500 | 100 | 200 | 270 |
| Set covering | Balas & Ho (1980) | Items Sets | 500 1000 | 1000 1000 | 2000 1000 | 100 | 130 | 160 |
| Maximum independent set | Bergman et al. (2016) | Nodes | 500 | 1000 | 1500 | 480 | 980 | 1480 |
| Multiple knapsack | Fukunaga (2011) | Items Knapsacks | 100 6 | 100 12 | 100 18 | 30 | 50 | 100 |

Table 3: Instance size for each benchmark. Performance is evaluated on test instances that match the size of the training instances, as well as on larger instances, to further assess the generalization capacity of our agents. Last three columns indicate the approximate number of integer variables after presolve, both for train (easy) and transfer (medium and hard) instances.

## B  TRAINING PIPELINE

**DQN Implementation**    In Algorithm 1, we provide a description of DQN-BBMDP training pipeline. Our DQN implementation includes several Rainbow-DQN features (Hessel et al., 2017): double DQN (Van Hasselt et al., 2016), $n$-step learning and prioritized experience replay (PER) Schaul (2015). Moreover, as DQN-BBMDP learns distributions representing $Q$-values, it integrates elements of Bellemare et al. (2017).

---

**Algorithm 1** DQN-BBMDP

---

**for** $t = 0...N - 1$ **do**

    Draw randomly an instance $P \sim p_0$.

    Solve $P$ by acting following a combined $\epsilon$-greedy and Boltzman exploration according to $\mathbf{q}_{\theta_t}$.

    Collect transitions along the generated tree $(s_i, a_i, \sum_{j=1}^{k} r_{i+j}, s_{i+k})$ and store them into a replay buffer $\mathcal{B}_{replay}$.

    Update $\theta_t$ using the loss described in (6) on transition batches drawn from $\mathcal{B}_{replay}$.

**end for**

---

**Exploration**    We train our agents following Boltzmann and $\epsilon$-greedy exploration combined. Concretely, agents select actions uniformly from $\mathcal{A}$ with probability $\epsilon$, while following a Boltzmann exploration strategy with temperature $\tau$ for the remaining probability $1 - \epsilon$. The decay rates for $\epsilon$ and $\tau$ are listed in Table 4.

| Module | Training parameter | Value |
|---|---|---|
| $Q$-learning | Batch size | 128 |
| | Optimizer | Adam |
| | Learning rate $l_r$ | $5 \times 10^{-5}$ |
| | Discount factor $\gamma$ | 1.0 |
| | Agent steps per network update | 10 |
| Replay buffer | Buffer minimum size $|\mathcal{B}_{replay}|_{init}$ | $20 \times 10^3$ |
| | Buffer maximum capacity $|\mathcal{B}_{replay}|_{max}$ | $100 \times 10^3$ |
| Prioritized experience replay | PER $\alpha$ | 0.6 |
| | PER $\beta_{init}$ | 0.4 |
| | PER $\beta_{final}$ | 1.0 |
| | $\beta_{init} \to \beta_{final}$ learner steps | $100 \times 10^3$ |
| | Minimum experience priority | $10^{-3}$ |
| | Soft target network update $\tau_{net}$ | $10^{-4}$ |
| | n-step DQN $k$ | 3 |
| Exploration | Start exploration probability $\epsilon_{init}$ | 1.0 |
| | Minimum exploration probability $\epsilon_{min}$ | $2.5 \times 10^{-2}$ |
| | $\epsilon$-decay | $10^{-4}$ |
| | Start temperature $\tau_{init}$ | 1.0 |
| | Minimum temperature $\tau_{min}$ | $10^{-3}$ |
| | $\tau$-decay | $10^{-5}$ |
| HL-Gauss (only for DQN-BBMDP) | $z_{min}$ | -1 |
| | $z_{max}$ | 16 |
| | $m_b$ | 18 |
| | $\sigma$ | 0.75 |

Table 4: Training parameters for all DQN branching agents. For DQN-Retro, we take $\gamma = 0.99$ as in Parsonson et al. (2022).

**Reward model** In section 3.1, we defined $\mathcal{R}(s, a) = -2$ for all transition, so that the overall value of a trajectory matched the size of the B&B tree. In practice, all negative constant reward model yield equivalent optimal policies in BBMDP, therefore, we chose to implement $\mathcal{R}(s, a) = -1$ for all RL baselines in order to allow clearer comparison between BBMDP and TreeMDP agents.

**Training parameters** Table 4 provides the list of hyperparameters used to train DQN agents on the Ecole benchmarks. To allow fair comparisons, when applicable, we keep SCIP parameters, training parameters and network architectures fixed for all DQN-agents.

## C NEURAL NETWORK

**State representation** Following the works of Gasse et al. (2019), MILPs are best represented by bipartite graphs $\mathcal{G} = (\mathcal{V}_\mathcal{G}, \mathcal{C}_\mathcal{G}, \mathcal{E}_\mathcal{G})$ where $\mathcal{V}_\mathcal{G}$ denotes the set of variable nodes, $\mathcal{C}_\mathcal{G}$ denotes the set of constraint nodes, and $\mathcal{E}_\mathcal{G}$ denotes the set of edges linking variable and constraints nodes. Nodes $v_\mathcal{G} \in \mathcal{V}_\mathcal{G}$ and $c_\mathcal{G} \in \mathcal{C}_\mathcal{G}$ are connected if the variable associated with $v_\mathcal{G}$ appears in the constraint associated with $c_\mathcal{G}$. Given a MILP $P$, defined as in section 2.1, its associate bipartite representation $\mathcal{G}$ has $|\mathcal{G}| = |\mathcal{V}_\mathcal{G}| + |\mathcal{C}_\mathcal{G}| = n + m$ nodes. We use bipartite graphs to represent $M \in \mathcal{M}$ as described in Section 3.2. In our experiments, IL and PG-tMDP agents use the list of features of Gasse et al. (2019) to represent variable nodes, constraint nodes

and edges, while DQN-BBMDP, DQN-TreeMDP and DQN-Retro agents also make use of the additional features introduced by Parsonson et al. (2022).

**Network architecture**  All RL agents utilize the graph convolutional network architecture described in Scavuzzo et al. (2022) and Parsonson et al. (2022). In DQN-BBMDP, the architecture differs slightly, with the final layer outputting distribution vectors in $\mathbb{R}^{m_b}$ instead of scalar values in $\mathbb{R}$.

## D  BASELINES

**Imitation learning**  We trained and tested IL agents using the official Ecole re-implementation of Gasse et al. (2019) shared at `https://github.com/ds4dm/learn2branch-ecole/tree/main`.

**DQN-TreeMDP**  Since there is no publicly available implementation of Etheve et al. (2020), we re-implemented DQN-TreeMDP and trained it on the four Ecole benchmarks, using when applicable the same network architectures and training parameters as in DQN-BBMDP and DQN-Retro. We share implementation and trained network weights to the community.

**PG-tMDP**  We used the official implementation of Scavuzzo et al. (2022) to evaluate PG-TreeMDP. For each benchmark, we used the tMDP+DFS network weights shared at `https://github.com/lascavana/rl2branch`.

**DQN-Retro**  As Parsonson et al. (2022) only trained on easy set covering instances, we took inspiration from the official implementation shared at `https://github.com/cwfparsonson/retro_branching` to train and evaluate DQN-Retro agents on the four Ecole benchmarks. Importantly, we trained and tested DQN-Retro following a BeFS node selection strategy, see Appendix F for more details. We share our re-implementation and trained network weights with the community.

## E  HL-GAUSS LOSS

As they investigated the uneven success met by complex neural network architectures such as Transformers in supervised versus reinforcement learning, Farebrother et al. (2024) found that training agents using a cross-entropy classification objective significantly improved the performance and scalability of value-based RL methods. However, replacing mean squared error regression with cross-entropy classification requires methods to transform scalars into distributions and distributions into scalars. Farebrother et al. (2024) found the Histogram Gaussian loss (HL-Gauss) (Imani & White, 2018), which exploits the ordinal structure structure of the regression task by distributing probability mass on multiple neighboring histogram bins, to be a reliable solution across multiple RL benchmarks. Concretely, in HL-Gauss, the support of the value function $\mathcal{Z} \subset \mathbb{R}$ is divided in $m_b$ bins of equal width forming a partition of $\mathcal{Z}$. Bins are centered at $z_i \in \mathcal{Z}$ for $1 \leq i \leq m_b$, we use $\eta = (z_{max} - z_{min})/m_b$ to denote their width. Given a scalar $z \in \mathcal{Z}$, we define the random variable $Y_z \sim \mathcal{N}(\mu = z, \sigma^2)$ and note respectively $\phi_{Y_z}$ and $\Phi_{Y_z}$ its associate probability density and cumulative distribution function. $z$ can then be encoded into a histogram distribution on $\mathcal{Z}$ using the function $p_{hist} : \mathbb{R} \to [0,1]^{m_b}$. Explicitly, $p_{hist}$ computes the aggregated mass of $\phi_{Y_z}$ on each bin:

$$p_{hist}(z) = (p_i(z))_{1 \leq i \leq m_b} \text{ with } p_i(z) = \int_{z_i - \frac{\eta}{2}}^{z_i + \frac{\eta}{2}} \phi_{Y_z}(y)dy = \Phi_{Y_z}(z_i + \frac{\eta}{2}) - \Phi_{Y_z}(z_i - \frac{\eta}{2})$$

Conversely, histogram distributions $(p_i)_{1 \leq i \leq m_b}$ such as the ones outputted by agents' value networks can be converted to scalar simply by computing the expectation: $z = \sum_{i=1}^{m_b} p_i \cdot z_i$.

BBMDP is a challenging setting to adapt HL-Gauss, as the support for value functions spans over several order of magnitude. In practice, we observe that for easy instances of the Ecole benchmark, $\mathcal{Z} = [-10^6, -2]$. Since value functions predict the number of node of binary trees built with B&B, it seems natural to choose bins centered at $z_i = -2^i$ to partition $\mathcal{Z}$. In order to preserve bins of equal size, we consider distributions on the support $\psi(\mathcal{Z})$ with $\psi(z) = \log_2(-z)$ for $z \in \mathcal{Z}$, such that $\psi(\mathcal{Z})$ is efficiently partitioned by bins centered at $z_i = i$ for $1 \leq i \leq m_b$. Thus, in BBMDP histograms distributions are given by $p_{hist}(z) = (p_i \circ \psi(z))_{1 \leq i \leq m_b}$ for $z \in \mathcal{Z}$, and can be converted back to $\mathcal{Z}$ through $z = \sum_{i=1}^{m_b} p_i \cdot \psi^{-1}(z_i)$ with $\psi^{-1}(z) = -2^z$.

## F    BBMDP vs Retro branching

In their work, Parsonson et al. (2022) proposed to train RL agents on retrospective trajectories built from TreeMDP episodes, in order to leverage the state-of-the art node selection policies implemented in MILP solvers. When reproducing their work, we found several discrepancies with the results they stated. First, the performance gap between DQN-Retro and DQN-TreeMDP (Etheve et al., 2020) turned out to be much narrower than expected. On easy set covering instances, the only benchmark on which the two agents are compared in Parsonson et al. (2022), we even found DQN-TreeMDP to perform better. Second, Parsonson et al. (2022) found that adopting a best-first-search (BeFS) node selection strategy at evaluation time greatly improved the performance of DQN-TreeMDP on easy set covering instances, indicating that abandoning DFS during training could be beneficial. However, in our experiments, we observed a 20% performance drop when replacing DFS by BeFS at evaluation time. After thorough examination of both Parsonson et al. (2022)'s article and implementation, we found that the baseline labeled as DQN-TreeMDP (FMSTS-DFS in their article) was quite distant from the branching agent originally described in Etheve et al. (2020). In fact, in Parsonson et al. (2022), the Etheve et al. (2020) branching agent is not trained on TreeMDP trajectories, but on retrospective trajectories built from TreeMDP episodes, using a DFS construction heuristic. Therefore, Parsonson et al. (2022) could not conclude on the superiority of retro branching over TreeMDP, nor could they assess the limitations of DFS-based RL agents. In contrast, our contribution provides compelling evidence that, while DFS is generally expected to hinder the training performance of RL agents due to its reputation as a suboptimal node selection policy, the theoretical guarantees brought by DFS in BBMDP enable to surpass prior state-of-the-art non-DFS agents. We believe this is because optimizing the node selection policy has less influence on tree size performance compared to optimizing the variable selection policy, as evidenced by Etheve (2021).

# G  VALIDATION CURVES

We trained our agents on one GPU NVIDIA A100 with 40GB of VRAM. We present validation curves for DQN-BBMDP, DQN-TreeMDP and DQN-Retro in Figure 4. For each benchmark we trained for 200k gradient steps, which took approximately 2 days for combinatorial auction instances, 3 days for set covering instances, 5 days for multiple knapsack instances and 7 days for maximum independent set instances. As shown in Figure 4, DQN-BBMDP training was interrupted before final convergence on 3 out of 4 benchmarks, hinting that performance could likely be improved by training for more steps.

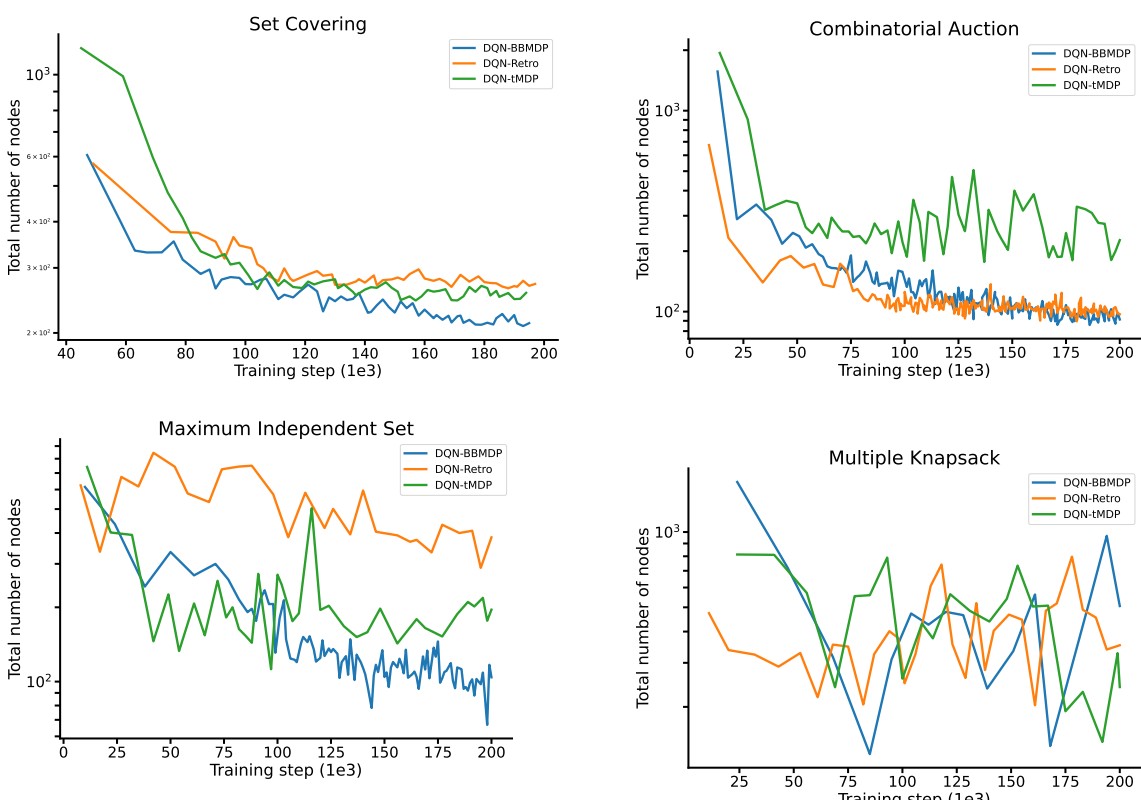

Figure 4: Validation curves for DQN-BBMDP, DQN-Retro and DQN-tMDP agents, in log scale. Throughout training, agents are evaluated on 20 validation instances after each batch of 100 training instances solved. Note that on the multiple knapsack benchmark, none of the agents reach convergence.

# H   FURTHER COMPUTATIONAL RESULTS

In this section, we include further computational results on instances of the Ecole benchmark. Table 5 provides computational results for harder instance benchmarks, as defined in Appendix A. Hard instances are solved within a time limit of 10 minutes. Since most instances cannot be solved within the time limit, we report final gaps and primal dual integrals averaged over 100 instances. The gap $g$ is defined as the normalized difference between (primal) global upper bound $GUB$ and (dual) global lower bound $GLB$ such that:

$$g = \frac{|GUB - GLB|}{\min(|GUB|, |GLB|)},$$

while the primal-dual integral is defined as the integral of $g$ with respect to running time. Among RL agents, DQN-Retro achieves the best performance across the four hard instance benchmarks, despite exhibiting the worst performance across the four medium instance benchmarks, as show in Table 1. We believe this to be due to the use of final gap and primal-dual integral as performance metrics. In fact, contrary to other RL agents, DQN-Retro is not trained to minimize B&B tree size. Instead, DQN-Retro learns variable selection strategies yielding the shortest diving trajectories, hence strategies favoring primal-dual gap reduction over tree size minimization. As a result, the DQN-Retro agent manages to achieve the best final gap and primal dual integral performance in average, all while solving fewer instances to optimality than other RL agents on hard instance benchmarks, as illustrated in Table 6. More importantly, we observe that while the IL expert demonstrates reasonable generalization capability, achieving performance comparable to SCIP on hard instance benchmarks, all RL baselines perform poorly, with none managing to outperform the random agent across the four benchmarks. This underscores the limited generalization capacity of current model-free RL agents to higher-dimensional instances, and highlights the need to adapt model-based RL approaches to the B&B setting, by leveraging the BBMDP framework. Indeed, model-based MCTS RL approaches have previously demonstrated the ability to surpass human expertise in combinatorial tasks such as board games (Schrittwieser et al., 2020).

Table 6 provides additional performance metrics to compare the different baselines across easy, medium and hard instance benchmarks. For each benchmark, we report the number of wins and the average rank of each baseline across 100 evaluation instances. The number of wins is defined as the number of instances where a baseline solves a MILP problem faster than any other baseline. When multiple baselines fail to solve an instance to optimality within the time limit, their performance is ranked based on final dual gap.

Finally, Table 7 recapitulates the computational results presented in Table 1, and provides for each baseline the per-benchmark standard deviation over five seeds, as well as the fraction of test instances solved to optimality within the time limit.

| Method | Set Covering | | Comb. Auction | | Max. Ind. Set | | Mult. Knapsack | | Norm. Score | |
|---|---|---|---|---|---|---|---|---|---|---|
| | Gap | Integral | Gap | Integral | Gap | Integral | Gap | Integral | Gap | Integral |
| Random | 12.5% | 30221 | 1.53% | 263389 | 2.80% | 12148 | **0.019%** | **485** | 65 | 50 |
| SB | 16.4% | 38691 | 1.90% | 342652 | 6.93% | 29712 | 0.277% | 2978 | 215 | 122 |
| RPB | 5.1% | 17229 | **0.07%** | **15687** | 2.25% | 10198 | 0.017% | 822 | **29** | **30** |
| IL | **4.9%** | **16997** | 0.13% | 62793 | **1.66%** | **8769** | 0.052% | 1666 | 43 | 48 |
| IL-DFS | 14.2% | 35167 | 0.63% | 183110 | 3.35% | 16282 | 0.133% | 2428 | 109 | 84 |
| PG-tMDP | 19.6% | **40566** | 1.90% | 327659 | **2.51%** | 12810 | 0.098% | 2518 | 113 | 93 |
| DQN-tMDP | 18.8% | 307117 | 2.60% | 392797 | 3.77% | 15701 | **0.019%** | **1021** | 95 | 98 |
| DQN-Retro | **10.0%** | 189758 | **1.58%** | **225552** | 2.82% | **12305** | 0.049% | 1285 | **76** | **74** |
| DQN-BBMDP | 18.5% | 303313 | 1.80% | 308803 | 3.75% | 16670 | 0.056% | 1452 | 100 | 100 |

Hard instances (Test)

Table 5: Computational results on hard instance benchmarks. For each method, we report the final gap as well as the primal-dual integral at the end of the solving time, averaged over 100 instances. Lower is better, red indicates best agent overall, blue indicates best among RL agents. Norm. Score denotes the aggregate average performance obtained by each agent across the four MILP benchmarks, normalized by the score of DQN-BBMDP.

| Method | Easy Solved | Easy Wins | Easy Rank | Medium Solved | Medium Wins | Medium Rank | Hard Solved | Hard Wins | Hard Rank |
|---|---|---|---|---|---|---|---|---|---|
| RPB | 100/100 | 8/100 | 5.7 | 100/100 | 10/100 | 3.3 | 27/100 | 35/100 | 1.83 |
| IL | 100/100 | 1/00 | 3.8 | 100/100 | 29/100 | 1.8 | 27/100 | **49/100** | **1.54** |
| IL-DFS | 100/100 | 35/100 | 2.1 | **100/100** | **58/100** | **1.7** | **29/100** | 16/100 | 3.75 |
| PG-tMDP | 100/100 | 0/100 | 6.6 | 78/100 | 0/100 | 6.8 | 1/100 | 0/100 | 6.34 |
| DQN-tMDP | 100/100 | 11/100 | 2.9 | 96/100 | 0/100 | 5.0 | 5/100 | 0/100 | 5.84 |
| DQN-Retro | 100/100 | 1/100 | 4.9 | 98/100 | 0/100 | 5.1 | 6/100 | 0/100 | **3.37** |
| DQN-BBMDP | 100/100 | **44/100** | **1.9** | **100/100** | **3/100** | **4.2** | **8/100** | 0/100 | 5.34 |

### Set covering

| Method | Easy Solved | Easy Wins | Easy Rank | Medium Solved | Medium Wins | Medium Rank | Hard Solved | Hard Wins | Hard Rank |
|---|---|---|---|---|---|---|---|---|---|
| RPB | 100/100 | 14/100 | 6.2 | 100/100 | 14/100 | 3.51 | **90/100** | **76/100** | **1.28** |
| IL | 100/100 | 3/100 | 3.5 | **100/100** | **47/100** | **1.7** | 80/100 | 20/100 | 1.94 |
| IL-DFS | 100/100 | 9/100 | 2.6 | 100/100 | 20/100 | 2.5 | 71/100 | 4/100 | 3.18 |
| PG-tMDP | 100/100 | 0/100 | 5.2 | 100/100 | 0/100 | 5.8 | **29/100** | 0/100 | 5.56 |
| DQN-tMDP | 100/100 | 0/100 | 5.5 | 100/100 | 0/100 | 6.5 | 2/100 | 0/100 | 6.58 |
| DQN-Retro | 100/100 | 6/100 | 3.6 | 100/100 | 1/100 | 4.6 | 27/100 | 0/100 | **4.00** |
| DQN-BBMDP | 100/100 | **74/100** | **1.5** | **100/100** | **18/100** | **2.9** | 21/100 | 0/100 | 5.47 |

### Combinatorial Auction

| Method | Easy Solved | Easy Wins | Easy Rank | Medium Solved | Medium Wins | Medium Rank | Hard Solved | Hard Wins | Hard Rank |
|---|---|---|---|---|---|---|---|---|---|
| RPB | 100/100 | 9/100 | 5.7 | 100/100 | 7/100 | 4.5 | 9/100 | 6/100 | 2.56 |
| IL | 100/100 | **72/100** | **1.6** | 100/100 | 36/100 | **1.7** | 20/100 | **70/100** | **1.31** |
| IL-DFS | 100/100 | 10/100 | 2.4 | 100/100 | 0/100 | 3.2 | 19/100 | 0/100 | 5.17 |
| PG-tMDP | 100/100 | 0/100 | 4.9 | **100/100** | **57/100** | **1.8** | **29/100** | **24/100** | **3.17** |
| DQN-tMDP | 100/100 | 1/100 | 4.8 | 85/100 | 0/100 | 6.1 | 0/100 | 0/100 | 6.01 |
| DQN-Retro | 100/100 | **6/100** | 5.4 | 22/100 | 0/100 | 6.7 | 0/100 | 0/100 | 3.56 |
| DQN-BBMDP | 100/100 | 2/100 | **3.3** | 95/100 | 0/100 | 4.2 | 1/100 | 0/100 | 6.22 |

### Maximum Independent Set

| Method | Easy Solved | Easy Wins | Easy Rank | Medium Solved | Medium Wins | Medium Rank | Hard Solved | Hard Wins | Hard Rank |
|---|---|---|---|---|---|---|---|---|---|
| RPB | 100/100 | **88/100** | **1.4** | **100/100** | **60/100** | **1.9** | 91/100 | **53/100** | **2.21** |
| IL | 100/100 | 1/100 | 4.5 | 100/100 | 6/100 | 3.4 | 86/100 | 15/100 | 3.78 |
| IL-DFS | 100/100 | 1/100 | 5.8 | 98/100 | 0/100 | 6.0 | 65/100 | 1/100 | 5.19 |
| PG-tMDP | 100/100 | 0/100 | 6.0 | 98/100 | 5/100 | 5.0 | 67/100 | 3/100 | 4.98 |
| DQN-tMDP | 100/100 | 1/100 | 3.5 | 99/100 | **14/100** | **3.5** | **91/100** | **14/100** | **3.32** |
| DQN-Retro | 100/100 | 3/100 | 3.5 | 98/100 | 9/100 | 3.8 | 72/100 | 6/100 | 4.15 |
| DQN-BBMDP | 100/100 | **6/100** | **3.3** | **100/100** | 6/100 | 4.3 | 75/100 | 8/100 | 4.37 |

### Multiple Knapsack

Table 6: Additional performance metrics for each baseline on easy, medium and hard instance benchmarks, see Appendix A for instance details. For each benchmark, we report the number of wins, and the average rank of each baseline across the 100 evaluation instances. We also report for each baseline the fraction of test instances solved to optimality within time limit. The number of wins is defined as the number of instances where a baseline solves a MILP problem faster than all other baselines. When multiple baselines fail to solve an instance to optimality within time limit, their performance is ranked based on final dual gap.

| Method | Easy Nodes | Time | Solved | Medium Nodes | Time | Solved |
|---|---|---|---|---|---|---|
| Random | $3289 \pm 4.2\%$ | $5.9 \pm 4.3\%$ | 100/100 | $270365 \pm 9.5\%$ | $811 \pm 7.9\%$ | 60/100 |
| SB | $35.8 \pm 0.0\%$ | $12.93 \pm 0.0\%$ | 100/100 | $672.1 \pm 0.0\%$ | $398 \pm 0.2\%$ | 82/100 |
| RPB | $62.0 \pm 0.0\%$ | $2.27 \pm 0.0\%$ | 100/100 | $3309 \pm 0.0\%$ | $48.4 \pm 0.1\%$ | 100/100 |
| IL | $133.8 \pm 1.0\%$ | $0.90 \pm 4.8\%$ | 100/100 | $2610 \pm 0.7\%$ | $23.1 \pm 1.5\%$ | 100/100 |
| IL-DFS | $136.4 \pm 1.8\%$ | $0.74 \pm 5.3\%$ | 100/100 | $3103 \pm 2.0\%$ | $22.5 \pm 3.1\%$ | 100/100 |
| PG-tMDP | $649.4 \pm 0.7\%$ | $2.32 \pm 2.4\%$ | 100/100 | $44649 \pm 3.7\%$ | $221 \pm 4.1\%$ | 78/100 |
| DQN-tMDP | $175.8 \pm 1.1\%$ | $0.83 \pm 4.5\%$ | 100/100 | $8632 \pm 4.9\%$ | $71.3 \pm 5.8\%$ | 96/100 |
| DQN-Retro | $183.0 \pm 1.2\%$ | $1.14 \pm 4.1\%$ | 100/100 | $6100 \pm 4.2\%$ | $59.4 \pm 4.2\%$ | 98/100 |
| DQN-BBMDP | $152.3 \pm 0.6\%$ | $0.77 \pm 5.6\%$ | 100/100 | $5651 \pm 2.2\%$ | $46.4 \pm 3.3\%$ | 100/100 |

### Set covering

| Method | Easy Nodes | Time | Solved | Medium Nodes | Time | Solved |
|---|---|---|---|---|---|---|
| Random | $1111 \pm 4.3\%$ | $2.16 \pm 6.6\%$ | 100/100 | $354650 \pm 6.7\%$ | $814 \pm 7.1\%$ | 64/100 |
| SB | $28.2 \pm 0.0\%$ | $6.21 \pm 0.1\%$ | 100/100 | $389.6 \pm 0.0\%$ | $255 \pm 0.2\%$ | 88/100 |
| RPB | $20.2 \pm 0.0\%$ | $1.77 \pm 0.1\%$ | 100/100 | $1376 \pm 0.0\%$ | $14.77 \pm 0.1\%$ | 100/100 |
| IL | $83.6 \pm 0.8\%$ | $0.73 \pm 7.3\%$ | 100/100 | $1309 \pm 1.6\%$ | $9.8 \pm 2.2\%$ | 100/100 |
| IL-DFS | $95.5 \pm 0.9\%$ | $0.67 \pm 7.3\%$ | 100/100 | $1802 \pm 2.0\%$ | $11.1 \pm 1.8\%$ | 100/100 |
| PG-tMDP | $168.0 \pm 2.8\%$ | $0.94 \pm 6.0\%$ | 100/100 | $6001 \pm 2.7\%$ | $30.7 \pm 2.4\%$ | 100/100 |
| DQN-tMDP | $203.3 \pm 4.2\%$ | $1.11 \pm 4.0\%$ | 100/100 | $20553 \pm 3.8\%$ | $116 \pm 3.9\%$ | 100/100 |
| DQN-Retro | $103.2 \pm 1.2\%$ | $0.78 \pm 7.5\%$ | 100/100 | $2908 \pm 1.7\%$ | $18.4 \pm 2.7\%$ | 100/100 |
| DQN-BBMDP | $97.9 \pm 1.2\%$ | $0.62 \pm 8.5\%$ | 100/100 | $2273 \pm 1.9\%$ | $11.8 \pm 2.0\%$ | 100/100 |

### Combinatorial auction

| Method | Easy Nodes | Time | Solved | Medium Nodes | Time | Solved |
|---|---|---|---|---|---|---|
| Random | $386.8 \pm 5.4\%$ | $2.01 \pm 4.8\%$ | 100/100 | $215879 \pm 6.7\%$ | $2102 \pm 6.2\%$ | 25/100 |
| SB | $24.9 \pm 0.0\%$ | $45.87 \pm 0.4\%$ | 100/100 | $169.9 \pm 0.2\%$ | $2172 \pm 0.9\%$ | 15/100 |
| RPB | $19.5 \pm 0.0\%$ | $2.44 \pm 0.4\%$ | 100/100 | $3368 \pm 0.0\%$ | $90.0 \pm 0.2\%$ | 100/100 |
| IL | $40.1 \pm 3.45\%$ | $0.44 \pm 3.1\%$ | 100/100 | $1882 \pm 4.0\%$ | $37.6 \pm 3.2\%$ | 100/100 |
| IL-DFS | $69.4 \pm 6.5\%$ | $0.56 \pm 4.8\%$ | 100/100 | $3501 \pm 2.7\%$ | $55.5 \pm 2.6\%$ | 100/100 |
| PG-tMDP | $153.6 \pm 5.0\%$ | $0.92 \pm 2.6\%$ | 100/100 | $3133 \pm 4.6\%$ | $39.5 \pm 3.8\%$ | 100/100 |
| DQN-tMDP | $168.0 \pm 5.6\%$ | $1.00 \pm 3.4\%$ | 100/100 | $45634 \pm 7.4\%$ | $477 \pm 5.1\%$ | 85/100 |
| DQN-Retro | $223.0 \pm 4.1\%$ | $1.81 \pm 3.6\%$ | 100/100 | $119478 \pm 6.1\%$ | $1863 \pm 4.8\%$ | 22/100 |
| DQN-BBMDP | $103.2 \pm 9.3\%$ | $0.69 \pm 6.8\%$ | 100/100 | $7168 \pm 5.3\%$ | $81.3 \pm 4.2\%$ | 95/100 |

### Maximum independent set

| Method | Easy Nodes | Time | Solved | Medium Nodes | Time | Solved |
|---|---|---|---|---|---|---|
| Random | $733.5 \pm 13.0\%$ | $0.55 \pm 6.9\%$ | 100/100 | $93452 \pm 14.3\%$ | $70.6 \pm 9.2\%$ | 99/100 |
| SB | $161.7 \pm 0.0\%$ | $0.69 \pm 0.1\%$ | 100/100 | $1709 \pm 0.5\%$ | $12.5 \pm 0.9\%$ | 100/100 |
| RPB | $289.5 \pm 0.0\%$ | $0.53 \pm 0.2\%$ | 100/100 | $30260 \pm 0.0\%$ | $22.14 \pm 0.2\%$ | 100/100 |
| IL | $272.0 \pm 12.9\%$ | $1.02 \pm 8.5\%$ | 100/100 | $9747 \pm 7.5\%$ | $46.5 \pm 6.6\%$ | 100/100 |
| IL-DFS | $472.8 \pm 13.0\%$ | $1.54 \pm 9.0\%$ | 100/100 | $43224 \pm 9.0\%$ | $177 \pm 8.6\%$ | 98/100 |
| PG-tMDP | $436.9 \pm 21.2\%$ | $1.57 \pm 16.9\%$ | 100/100 | $35614 \pm 14.3\%$ | $165 \pm 15.4\%$ | 98/100 |
| DQN-tMDP | $266.4 \pm 7.2\%$ | $0.73 \pm 4.6\%$ | 100/100 | $22631 \pm 8.6\%$ | $65.1 \pm 5.5\%$ | 99/100 |
| DQN-Retro | $250.3 \pm 9.5\%$ | $0.67 \pm 5.0\%$ | 100/100 | $27077 \pm 8.8\%$ | $79.5 \pm 6.2\%$ | 98/100 |
| DQN-BBMDP | $236.6 \pm 6.4\%$ | $0.66 \pm 2.7\%$ | 100/100 | $37098 \pm 7.0\%$ | $109 \pm 4.9\%$ | 100/100 |

### Multiple knapsack

Table 7: Computational performance comparison on four MILP benchmarks. Following prior works, we report geometrical mean over 100 instances, averaged over 5 seeds, as well as per-benchmark standard deviations.