# OpenReview forum: "A Markov decision process for variable selection in Branch and bound"
_ICLR.cc/2025/Conference — Submitted to ICLR 2025_

### Official Review · Reviewer_jMj7 · 2024-11-04

**Soundness:** 2
**Presentation:** 3
**Contribution:** 2
**Rating:** 5
**Confidence:** 4

**Summary:**

This work addresses the issue that these alternative MDP formulations introduce approximations that undermine the asymptotic performance of RL branching agents in the general case. This paper introduces a B&B MDP formulation for variable selection in B&B, which preserves optimality without sacrificing the convergence properties brought by previous contributions. Computational experiments validate our model empirically, as our branching agent outperforms prior state-of-the-art RL agents on four standard MILP benchmarks.

**Strengths:**

1)	Original: This paper introduces a vanilla MDP formulation for variable selection in B&B, allowing the leveraging of a broad range of RL algorithms to learn optimal B&B heuristics.
2)	Quality:
The overall presentation is good
The paper discusses the related works about RL-based methods in a fairly clear manner.
3)	Clarity:
Overall, the paper is clear and easy to follow, however, some writing is confusing, especially since some symbols are not defined.
4)	Significance:
The paper studies introduce a vanilla MDP applied to the variable selection problem to learn optimal B&B branching strategies, which seems to be an important research direction.

**Weaknesses:**

1.	The related work about imitation learning for variable selection is insufficient,
2.	There is no pipeline to explain how the proposed method works.
3.	The proposed method seems to only apply to the binary integer programming problem, it is a little unclear to me how much technical novelty is in the B&B MDP and whether the contributions in this paper are significant enough.
4.	The experiment seems insufficient, only testing in easy and medium difficulty levels, lack of comparison in hard difficulty levels.
5.	The ideas in the paper took me some time to properly digest. I believe all of the information needed for the reader to digest is there, but think that the paper could make this process easier for the reader.

**Questions:**

1.	The meaning of symbols is confusing. For example, what’s the meaning of ($v, \varepsilon$) in Line 84?
2.	Why is the reward defined as -2 in the MDP definition, as I know, some papers define the reward as -1[1], can you explain it?
3.	This may be important because the convergence guarantees of RL algorithms are often made in the discounted setting with litter than 1, Why is this paper setting the discounted with 1. Is there any experimental or theoretical support for this point?
4.	According to Gasse et al. (2019), why was this paper not compared to the facilities dataset? Why was this paper not compared to the hard dataset? Why does this paper not report the average per-instance standard deviation in Table 1?
5.	Why did this paper only test 20 instances on the medium transfer instances instead of testing 100 instances like the easy difficulty level? What would happen if this method also tested the 100 instances on the medium difficulty level?
6.	For the variable selection problem, reinforcement learning seems to have no advantage over imitation learning both in training speed and testing effectiveness. So, what is the motivation behind our research in this paper?
7.	Some articles have already defined branching as MDP [1]. Can you summarize the differences between you and these articles?
8.	The contribution summary of the article is unclear. Can you further summarize it?
[1] Improving Learning to Branch via Reinforcement Learning. NeurIPS Workshop, 2020. https://openreview.net/forum?id=z4D7-PTxTb

---

> ### Author Response · Authors · 2024-11-22
> **Response to reviewer jMj7 (1/3)**
>
> We thank the reviewer for their valuable time and detailed feedback. We have made modifications to the paper that we hope improve clarity and presentation. We start by addressing the concerns raised by the reviewer in order.
>
> > The related work about imitation learning for variable selection is insufficient.
>
> 1. We thank the reviewer for raising this issue. At the reviewer’s request, we added a paragraph to our related work section line 153-159, which discusses extensions to the work of (Gasse et al. 2019) as well as theoretical limitations.
>
> > There is no pipeline to explain how the proposed method works.
>
> 2. We thank the reviewer for raising their concern on the absence of a general pipeline description. The overall DQN-BBMDP pipeline is actually simple, and very similar to the ones of DQN-retro (Parsonson et al. 2022) and (Scavuzzo et al. 2022), as described in section 4 line 338-341. Additionally, we provide the pseudocode for DQN-BBMDP as well as a full list of hyperparameters in Appendix B, along with a description of neural network architecture and state representation functions used to train RL agents are detailed in Appendix C. Is that inline with the reviewer’s expectations regarding pipeline description?
>
> > The proposed method seems to only apply to the binary integer programming problem, it is a little unclear to me how much technical novelty is in the B&B MDP and whether the contributions in this paper are significant enough.
>
> 3. We thank the reviewer for raising their concern, however, our work is not limited to binary problems. Throughout the document, we consider general mixed integer linear programs, as defined line 75-79. If the reviewer is interested in applications of BBMDP beyond the MILP framework, we stress that our method effectively generalizes to any combinatorial problem traditionally solved by branch and bound, meaning any combinatorial problem for which there exist efficient separation and evaluation methods.  Regarding the primary innovations brought by BBMDP, we added a general comment that we hope helps clarify the technical novelty of our contribution.
>
> >The experiment seems insufficient, only testing in easy and medium difficulty levels, lack of comparison in hard difficulty levels.
>
> 4. We thank the reviewer for raising this issue. Since no prior RL contribution tested its agents on higher-dimensional problems,  for the sake of comparison, we did not carry out tests on harder instance benchmarks at first. In fact, RL agents more than IL agents have been found to struggle to generalize to higher dimensions, which is why prior contributions only evaluated generalization on medium benchmarks. Nevertheless, we will include performance comparison between RL and IL agents on the hard benchmark shortly.
>
> > The ideas in the paper took me some time to properly digest. I believe all of the information needed for the reader to digest is there, but think that the paper could make this process easier for the reader.
>
> 5. We have made several revisions to the paper to clarify our contribution. We hope these changes enhance the readability and make the paper more digestible for the reader.

---

> ### Author Response · Authors · 2024-11-22
> **Response to reviewer jMj7 (2/3)**
>
> We now address the questions explicitly raised by the reviewer.
>
> > The meaning of symbols is confusing. For example, what’s the meaning of V, E in Line 84?
> 1. We thank the reviewer for sharing their confusion regarding notations. We adopted a standard convention in graph theory, noting V the set of vertices and E the set of edges. Please indicate whether there are further disturbing notations.
>
> > Why is the reward defined as -2 in the MDP definition, as I know, some papers define the reward as -1, can you explain it?
>
> 2. We thank the reviewer for raising this question. In fact, defining r= -2, r=-1, or r = -10 yields equivalent optimal branching policies for BBDMP. We defined r = -2 so that W-values of node o_i could be interpreted as the size of the subtrees rooted in o_i, which simplifies explanations in section 3.2. However, in our experiments, we implemented r = -1 for all agents in order to allow clearer comparison. We rewrote section 3.1, line 212-216 as well as Appendix B in order to make this explicit.
>
> >This may be important because the convergence guarantees of RL algorithms are often made in the discounted setting with litter than 1, Why is this paper setting the discounted with 1. Is there any experimental or theoretical support for this point?
>
> 3. We thank the reviewer for raising this issue. Like (Etheve et al. 2020) and (Scavuzzo et al. 2022),  in order to ensure alignment with the objective described in Equation 1 line 131, we set the discount factor to γ=1. This is made possible by the finite length of BBMDP episodes. Setting γ=1 also simplifies Equation (3) in Proposition 1, which, in turn, streamlines the implementation process.
>
> > According to Gasse et al. (2019), why was this paper not compared to the facilities dataset? Why was this paper not compared to the hard dataset? Why does this paper not report the average per-instance standard deviation in Table 1?
>
> 4. We thank the reviewer for raising their concern on the absence of the Capacity Facility Location benchmark, which is present in (Gasse et al. 2019) and (Scavuzzo et al. 2022). In fact, (Gasse et al. 2019) and (Scavuzzo et al. 2022) used former versions of SCIP, for which these instances were challenging to solve; this is not the case anymore. Owing to improvements in SCIP 8.0.1 presolve module, Capacity Facility Location instances as described in (Scavuzzo et al. 2022) are now solved in typically 2 to 5 fives nodes, when they are not solved directly to optimality during presolve, rendering training on this benchmark trivial.  Regarding the second and third part o the reiewer's question, we will include harder datasets as well as the per-benchmark standard deviation in the Appendix.
>
> > Why did this paper only test 20 instances on the medium transfer instances instead of testing 100 instances like the easy difficulty level? What would happen if this method also tested the 100 instances on the medium difficulty level?
>
> 5. We thank the reviewer for pointing this out. We include computational results averaged over 100 medium instances, averaged over 5 seeds.
>
> > For the variable selection problem, reinforcement learning seems to have no advantage over imitation learning both in training speed and testing effectiveness. So, what is the motivation behind our research in this paper?
>
> 6. We thank the reviewer for raising this issue. As discussed in section 1 line 47, we know that if IL yields better performance at the moment, its performance is capped by that of strong branching, which is not an optimal expert for variable selection. Moreover, as now discussed in section 2.3 line 153-159, theoretical and computational studies have shown that even the suboptimal performance of SB was out of reach for neural networks trained by imitation. In fact, low tree sizes associated with SB happen to be primarily due to the formulation improvements resulting from the massive number of LP solved by SB, rather than to the intrinsic quality of branching decisions themselves. Therefore, replicating the exact branching decisions made by SB results in significantly larger trees compared to SB itself, further limiting the performance that IL agents can achieve.

---

> ### Author Response · Authors · 2024-11-22
> **Response to reviewer jMj7 (3/3)**
>
> > Some articles have already defined branching as MDP. Can you summarize the differences between you and these articles?
>
> 7. The article quoted by the reviewer trains RL agents with evolutionary strategies using a novelty score based on a Wasserstein distance to measure the similarity between current and new candidate policies. Building on the work of (He et al. 2014), just like (Etheve et al. 2020) and (Scavuzzo et al. 2022), they effectively define branching as an MDP resembling our BBMDP. Explicitly, the difference between the MDP described in (Sun et al. 2020) and BBMDP is that we include all the nodes of the B&B tree in the current state, while they only consider open nodes. This is because our bipartite graph representation of B&B nodes include statistics on the whole B&B tree, therefore, keeping track of the B&B tree is mandatory to preserve the Markov property.
> Importantly the MDP formulation described in (Sun et al. 2020), similar to the one labeled as temporal MDP in (Scavuzzo et al. 2022), has been found impractical by (Etheve et al. 2020) and (Scavuzzo et al. 2022) to learn efficient branching strategy with approximate dynamic programming and policy gradient algorithms, which is why they preferred using TreeMDP. Our contribution proposes a synthesis of the temporal MDP and TreeMDP frameworks described in (Scavuzzo et al. 2022), that preserves the training convergence properties brought by TreeMDP as well as fundamental MDP properties.
>
> > The contribution summary of the article is unclear. Can you further summarize it?
>
> 8. We thank the reviewer for allowing us to clarify our contribution. We partly rewrote the introduction section, see lines 50-57, to make our contribution more explicit.  In our work, we show that despite improving the convergence of RL algorithms, the TreeMDP formulation introduces approximations which undermine the asymptotic performance of RL branching agents in the general case. In order to address this issue, we introduced BBMDP, a principled vanilla MDP formulation for variable selection in B&B, which preserves convergence properties brought by TreeMDP without sacrificing optimality. Computational experiments validate our approach, as the DQN-BBMDP agent obtains both better training and generalization performance on the Ecole benchmark over TreeMDP agents. We believe that leveraging the BBMDP framework is essential for advancing learning-based variable selection strategies in the future, as it enables the integration of state-of-the-art MCTS model-based RL algorithms that have proven capable of surpassing human knowledge in combinatorial tasks.

---

> > ### Comment · Reviewer_jMj7 · 2024-11-25
> >
> > Thank you for answering my related questions. I still have the following questions.
> >
> > 1. The state is the set of all B&B trees, will this result in very slow inference time for the model?
> >
> > 2. Understand the difficulties of RL methods in the scale of hard problems you mentioned and look forward to the results of this paper on hard instances.
> >
> > 3. Because the paper only compares 0-1 integer programming problems, although the rebuttal period is limited, there are still concerns about its effectiveness in integer programming, mixed integer programming, and real-world datasets.

---

> > > ### Author Response · Authors · 2024-11-27
> > > **Response to Reviewer jMj7**
> > >
> > > We thank the reviewer for their reply, allowing us to further clarify our contribution. We adress the questions raised by the reviewer in order.
> > >
> > > > The state is the set of all B&B trees, will this result in very slow inference time for the model?
> > >
> > > 1.  This is true when considering BBMDP agents in a non-DFS setting. However, in DFS, since minimizing the size of the subtree associated with the current B&B node is equivalent to acting according to a policy minimizing the size of the whole B&B tree, BBMDP agents only evaluate the values associated with the branching decisions at the current B&B node, before taking a greedy step based on these values. Therefore, in BBMDP, the inference time is strictly equivalent to that in TreeMDP.
> > >
> > > > Understand the difficulties of RL methods in the scale of hard problems you mentioned and look forward to the results of this paper on hard instances.
> > >
> > > 2.  Computational results on hard instance benchmarks are now presented in Appendix H. Since most hard instances are not solved to optimality within the time limit, performance is evaluated based on the final optimality gap and the primal-dual integral. In order to better differentiate between baselines across easy, medium and hard instance benchmarks, we include additional performance metrics such as wins and average baseline rank, as shown in Table 6. We wrote a general comment that we hope helps clarify the additional computational experiments included in our last revised submission.
> > >
> > > > Because the paper only compares 0-1 integer programming problems, although the rebuttal period is limited, there are still concerns about its effectiveness in integer programming, mixed integer programming, and real-world datasets.
> > >
> > > 3.  We understand the concern raised by the reviewer.  We argue that, in mixed-integer linear programming, the solving efficiency does not stem from whether the variables are binary, integer or mixed-integer, but rather from the quality of problem formulations in terms of their impact on bounds, problem size, and algorithmic features such as cutting and branching strategies. In other terms, the difficulty of solving MILPs arises not from the nature of the variables but from the combinatorial structure of the problem. The Ecole benchmark is a well-established benchmark for training and evaluating MILP learning agents in general, beyond variable selection. Our experimental protocol is inline with that of prior IL and RL contributions, who did train and test their baselines exclusively on the Ecole benchmark. Notwithstanding, we agree that showcasing enhanced performance on varied real-world MILP instance datasets is critical for the development of learning-based branching strategies in the future. Since our method is not specifically tailored to binary problems, we are confident that it generalizes effectively to the broader mixed-integer linear programming setting. In fact, we agree with the reviewer that the Ecole benchmark could benefit from the inclusion of a more diverse set of problem instances, featuring integer and mixed-integer variables.

---

> ### Author Response · Authors · 2024-12-02
> **Follow up**
>
> Dear Reviewer jMj7,
>
> Thank you again for your valuable feedback. We have conducted additional experiments on medium and hard instances that we integrated to Appendix H, along with additional performance metrics. We also provided thorough discussions to address your concerns. As the rebuttal period is nearing its conclusion, we wanted to follow up to ensure our responses have adequately addressed the reviewers' concerns. Please let us know if there are any remaining questions or areas where further clarification is needed. We appreciate the time and effort you have dedicated to reviewing our work.
>
> Sincerely,
>
> Authors

---

### Official Review · Reviewer_3mcR · 2024-11-05

**Soundness:** 2
**Presentation:** 3
**Contribution:** 2
**Rating:** 5
**Confidence:** 4

**Summary:**

This paper introduces a novel Markov Decision Process (MDP) framework, BBMDP, for optimizing variable selection in Branch and Bound (B&B) algorithms in Mixed Integer Programming (MIP). By restricting the node selection strategy to depth-first search (DFS), the authors derive a more canonical Bellman equation for BBMDP, enabling broader use of reinforcement learning frameworks. This approach offers greater robustness than existing TreeMDP models by preserving optimality and convergence properties. The authors apply Deep Q-learning to BBMDP, demonstrating improved performance over previous RL approaches on TreeMDP across multiple MILP benchmarks, with significant reductions in both computation time and the number of B&B nodes required. Results further suggest that BBMDP effectively narrows the gap between reinforcement learning and imitation learning methods.

**Strengths:**

1. The paper is well-written and easy to follow
2. The introduction of BBMDP provides a principled, canonical MDP formulation for variable selection in B&B, addressing limitations of previous TreeMDP approaches by enabling a broader application of reinforcement learning techniques with theoretical support.
3. The paper includes experiments on many standard MILP benchmarks

**Weaknesses:**

The adoption of a DFS node selection strategy allows for a more canonical Bellman operator and potentially broader applicability of current RL frameworks. However, the impact of this restriction on performance is not fully explored, and the paper lacks a discussion on potential trade-offs related to this design choice.

The authors highlight BBMDP’s potential to support a wider range of RL algorithms, yet only DQN, which is also compatible with TreeMDP, is tested. Consequently, the empirical results do not demonstrate the benefits of BBMDP’s broader RL applicability.

For medium instance testing, only 20 instances are evaluated, which may be insufficient to reliably represent each problem class. Increasing the test set to at least 100 instances would provide a more robust assessment of performance across diverse scenarios.

When comparing methods on node count and solution time, there are inconsistencies between the two metrics: a shorter runtime does not always correspond to fewer nodes processed. For instance, DQN-BBMDP sometimes achieves a lower node count yet requires more time, and vice versa when compared with IL approaches.

While the proposed methods show improved performance and reduce the gap with IL approaches, a considerable performance disparity remains.

Additionally, all tested problem classes are synthetic mathematical benchmarks, leaving the performance on realistic datasets unexamined.

**Questions:**

Besides the concerns raised in the weakness part, I have the following additional questions:

1. Why define R(s, a) = −2 for all transitions until episode termination? Is intuition behind the number -2? Is that the results of tuning?
2. Any intuition of why choosing the HL-Gauss cross-entropy loss?
3. Why not including IL-DFS to the medium problem classes? Will the drop of the performance of using DFS will become more significant for the larger instances?

---

> ### Author Response · Authors · 2024-11-22
> **Response to reviewer 3mcR (1/2)**
>
> We thank the reviewer for their valuable time and detailed feedback. We have made modifications to the paper that we hope improve clarity and presentation. We address the concerns raised by the reviewer in order.
>
> > The adoption of a DFS node selection strategy allows for a more canonical Bellman operator and potentially broader applicability of current RL frameworks. However, the impact of this restriction on performance is not fully explored, and the paper lacks a discussion on potential trade-offs related to this design choice.
>
> We thank the reviewer for pointing out this issue. Ideally, variable and node selection policies should be optimized jointly in Equation (1) in order to fully optimize the performance of B&B over a distribution of instances. Following findings (Etheve 2021), who found node selection optimization to be negligible against variable selection optimization, we focus on variable selection, and adopt a DFS node selection policy to enable the decomposition of the V-value function into a sum of independent W-values. Incidentally, our DFS-based DQN-BBMDP agent was found to clearly outperform the prior state-of-the-art DQN-retro agent, which uses SCIP default node selection heuristic. Moreover, when training and testing DQN-BBMDP following this same SCIP node selection heuristic, we found performance to be strictly equivalent to the DFS baseline, which suggests that the adoption of DFS is not compulsory to maintain training performance. We believe this to be due to the behavior of the SCIP default node selection heuristic, which exhibits diving behavior resembling that of DFS.  This is all discussed in the ablation study in section 4.4 and in Appendix F, referred to line 453, that we partly rewrote to address more specifically the reviewer’s concern.
>
> > The authors highlight BBMDP’s potential to support a wider range of RL algorithms, yet only DQN, which is also compatible with TreeMDP, is tested. Consequently, the empirical results do not demonstrate the benefits of BBMDP’s broader RL applicability.
>
> We thank the reviewer for raising this issue. As highlighted in Figure 3, we show that BBMDP is better suited than TreeMDP for k-step temporal difference learning, since k-step DQN-TreeMDP yields inconsistent Bellman updates. Our computational experiments support this finding. As shown in Table 1 and 2, p. 9-10, DQN-BBMDP clearly outperforms DQN-tMDP as well as other TreeMDP agents, while the ablation study highlights that k-step temporal difference improves the performance of DQN-BBMDP agent, while it undermines the performance of Tree-DQN BBMDP.  Moreover, we believe that leveraging the BBMDP framework is essential for advancing learning-based variable selection strategies in the future, as it enables the integration of state-of-the-art MCTS model-based RL algorithms that have proven capable of surpassing human intelligence in combinatorial tasks.
>
> > For medium instance testing, only 20 instances are evaluated, which may be insufficient to reliably represent each problem class. Increasing the test set to at least 100 instances would provide a more robust assessment of performance across diverse scenarios.
>
> We thank the reviewer for pointing this out. We include computational results over 100 medium instances, averaged over 5 seeds, in our latest submission.
>
> > When comparing methods on node count and solution time, there are inconsistencies between the two metrics: a shorter runtime does not always correspond to fewer nodes processed. For instance, DQN-BBMDP sometimes achieves a lower node count yet requires more time, and vice versa when compared with IL approaches.
>
> We thank the reviewer for pointing this out. In fact, if time and node metrics are generally aligned in B&B (this is reflected overall in our results tables) this is not automatic. In fact, strategy A may yield more nodes and yet be faster than strategy B, for example if the nodes explored by strategy A require in average fewer simplex iterations to compute linear relaxations. Nevertheless, RL branching agents are generally trained to minimize B&B tree size, as this objective has proven to be a more robust and consistent target across various hardware configurations, thus fostering performance reproducibility.

---

> ### Author Response · Authors · 2024-11-22
> **Response to reviewer 3mcR (2/2)**
>
> > While the proposed methods show improved performance and reduce the gap with IL approaches, a considerable performance disparity remains.
>
>  We thank the reviewer for raising their concern.  In fact, we believe the performance gains presented in Table 1 are substantial. We include two additional columns in Table 1, showing the aggregate average performance obtained by each agent across the four MILP benchmarks, normalized by the score of DQN-BBMDP. It appears that the performance gap between IL and DQN-BBMDP is narrower than the gap between DQN-BBMDP and any other RL agent. This is particularly true when evaluating generalization performance on the medium dataset, as discussed line 435-437 in the latest version, which underpins the computational benefit of adopting BBDMP over TreeMDP.
>
> > Additionally, all tested problem classes are synthetic mathematical benchmarks, leaving the performance on realistic datasets unexamined.
>
> We thank the reviewer for raising their concern. We argue that the Ecole benchmark is a well-established benchmark for training and evaluating learning agents in general, beyond variable selection. Furthermore, our experimental protocol is in line with that of prior IL and RL contributions, who trained and tested their baselines exclusively on the Ecole benchmark. Notwithstanding, we agree with the reviewer that showcasing enhanced performance on real-world instance datasets is critical for the development of learning-based branching strategies in the future.
>
> We now address the question explicitly raised by the reviewer.
>
> > Why define R(s, a) = −2 for all transitions until episode termination? Is intuition behind the number -2? Is that the results of tuning?
>
> 1. We thank the reviewer for raising this question. In fact, defining r= -2, r=-1, or r = -10 yields equivalent optimal branching policies for BBDMP. We defined r = -2 so that W-values of node o_i could be interpreted as the size of the subtree rooted in o_i, which simplifies explanations in section 3.2. However, in our experiments, we implemented r = -1 for all agents in order to allow clearer comparison with prior RL baselines. We rewrote section 3.1, line 212-216 as well as Appendix B in order to make this explicit.
>
> > Any intuition of why choosing the HL-Gauss cross-entropy loss?
>
> 2. We thank the reviewer for raising this question. Complete description and theoretical motivation for the use of HL-Gauss loss can be found in Appendix E, referred to line 316.
>
> > Why not including IL-DFS to the medium problem classes? Will the drop of the performance of using DFS will become more significant for the larger instances?
>
> 3. We thank the reviewer for pointing this out. The performance gap between IL and IL-DFS on medium instances is inline with that on easy instances.  We include these results in the latest submission, in order to remove ambiguity.

---

> > ### Comment · Reviewer_3mcR · 2024-11-27
> >
> > Thank you for your detailed response! My primary concern for this paper is its performance, particularly when compared to IL-based methods. For medium-sized instances, the method tends to involve more nodes and results in longer solving times. While the performance gap with IL-based solvers is narrower than that of RL-based methods, the gaps themselves remain significant and warrant further improvement. Also, the introduction of DFS to IL generally decreases the performance of IL-based methods, therefore I am not fully convinced that the introduction of DFS is insignificant. Or could you provide explanations for why IL is sensitive to the choice of node selection strategy and the RL-based methods are not?

---

> > > ### Author Response · Authors · 2024-11-29
> > > **Response to reviewer 3mcR 1/3**
> > >
> > > We thank the reviewer for articulating their concerns so clearly and thoroughly. We understand that the reviewer has highlighted two specific issues, which we address in order.
> > >
> > > > My primary concern for this paper is its performance, particularly when compared to IL-based methods. For medium-sized instances, the method tends to involve more nodes and results in longer solving times. While the performance gap with IL-based solvers is narrower than that of RL-based methods, the gaps themselves remain significant and warrant further improvement.
> > >
> > > We agree with the reviewer that, presently, in generalization, the performance of RL agents fall short when compared with IL agents. In fact, our experiments show that while the BBMDP framework improves both training convergence and generalization performance of RL agents, traditional approximate value iteration methods still struggle to handle the combinatorial complexity of MILP problems (as evidenced by IL-DFS outperforming every RL baseline on medium and hard instance benchmarks). However, as outlined in lines 46-48, IL approaches are inherently limited by the performance of the expert they learn from. Moreover, as discussed in lines 155-159, strong branching is not a very good expert for learning to branch, which has led to a stagnation in IL performance over the past five years.
> > >
> > > We believe this highlights the rationale for leveraging the BBMDP framework to adapt MCTS model-based algorithms, which have demonstrated the ability to surpass human knowledge in complex combinatorial tasks, to the B&B setting. We are confident that such adaptations could allow RL approaches to surpass the performance of IL agents in the future.

---

> > > ### Author Response · Authors · 2024-11-29
> > > **Response to reviewer 3mcR 2/3**
> > >
> > > >  Also, the introduction of DFS to IL generally decreases the performance of IL-based methods, therefore I am not fully convinced that the introduction of DFS is insignificant. Or could you provide explanations for why IL is sensitive to the choice of node selection strategy and the RL-based methods are not?
> > >
> > > We thank the reviewer for raising this issue. We believe that we can provide a clear explanation. In fact, IL methods are not designed to learn optimal branching policies as defined by Equation (1). Instead, they focus on replicating the behavior of strong branching. Because strong branching decisions are agnostic to the global B&B optimization process and depend solely on the MILP associated with the current B&B node, IL methods can effectively learn to mimic strong branching by disregarding both the node selection policy and the broader B&B tree structure during training. Consequently, when branching according to SB (or IL agents trained to imitate SB), using DFS instead of SCIP’s default node selection policy tends to reduce B&B solving efficiency, as exemplified in Table 1. This aligns with the general understanding that DFS is not typically an efficient node selection strategy.
> > >
> > > In contrast, the reinforcement learning (RL) approach aims to learn optimal branching strategies tailored to specific instance distributions, which, as defined in section 2.1, are necessarily defined with respect to a node selection policy. This naturally raises an important question, as noted by the reviewer: how does the optimal branching policy associated with a DFS node selection policy compare to optimal branching policies associated with state-of-the-art node selection strategies, such as those implemented in modern MILP solvers? Unfortunately, providing a direct quantitative answer to this question is not feasible, as true optimal branching policies are computationally intractable to compute, even for relatively simple instances. Yet, as the reviewer suggested, we believe that, in general, the optimal branching strategies associated with DFS are likely to produce larger B&B trees compared to those associated with SCIP’s default node selection strategy.
> > >
> > > However, this is not the only factor to consider when reflecting on which policy to adopt for training RL agents to learn optimal branching strategies. In fact, contrary to the IL setting, in RL the value of the current B&B state is not simply a function of the current B&B node, but a function of the whole B&B tree. As a result, in RL, the bipartite graph state representation and the associated GNN architecture introduced by Gasse et al. (2019) cannot be directly used to learn optimal value functions, as they fail to capture the information from other nodes of the B&B tree. Previous works have explored various solutions to address this issue. Like Etheve et al. (2019) and Scavuzzo et al. (2022), we opted for adopting a DFS node selection strategy, which allows to decompose the value of B&B states into the values of the independent subtrees rooted in the open nodes of the B&B tree. As shown in section 3.2 and 3.3, this allows to use both bipartite graph representations and traditional Q-learning algorithm to train GNNs to approximate the optimal value function (and, in turn, the optimal branching strategy) corresponding to the DFS node selection strategy. Other works, such as Parsonson et al. (2022), have chosen to use bipartite graph state representations while retaining SCIP’s default node selection policy, despite the approximations introduced in their learning scheme, in order to enhance performance.
> > >
> > > In our experiments, we found DQN-BBMDP to outperform DQN-Retro both on easy and medium instance benchmarks, suggesting that the benefits of adopting DFS outweigh those of adopting a more efficient node selection policy. More specifically, when learning efficient variable selection strategies, our experiments show that adopting a suboptimal node selection strategy that enhances training convergence properties yields better performance than using an efficient node selection policy while relying on approximate training updates to optimize the branching policy. This is consistent with the findings of Etheve et al. (2021), who observed that the impact of optimizing the node selection policy is negligible compared to the effects of optimizing the variable selection policy.

---

> > > ### Author Response · Authors · 2024-11-29
> > > **Response to reviewer 3mcR 3/3**
> > >
> > > To answer the reviewer’s question, we believe the reason why RL methods are less sensitive than IL methods to the use of DFS directly stems from the trade-offs discussed above. As a side note, we highlight that, by considering the entire B&B tree, the BBMDP framework also allows for the joint optimization of node and variable selection strategies. In this case, action selection involves choosing both an open node and an integer variable to branch on at each iteration. Such a reformulation is not feasible within the TreeMDP framework, as states are defined as current B&B nodes, which, by definition, are determined by the node selection policy and thus cannot be governed by the action selection policy. Hence, BBMDP framework appears as a necessary step towards the joint optimization of node and variable selection strategies in the future. Currently, this approach faces two major challenges. First, defining a representation function for the entire B&B tree, which involves dealing with trees where each node is itself a bipartite graph. Second, designing an efficient function approximator for such a complex data structure, particularly in addressing the information redundancy that leads to significant computational overhead.
> > >
> > > We hope our response has clarified the trade-offs involved in selecting a node selection policy within the context of learning efficient branching strategies through reinforcement learning. We remain available to address any further questions or concerns the reviewer may have.

---

> ### Author Response · Authors · 2024-12-02
> **Follow up**
>
> Dear Reviewer 3mcR,
>
> Thank you again for your valuable feedback. We have conducted further experiments and provided thorough discussions to address your orignal concerns. As the rebuttal period is nearing its conclusion, we wanted to follow up to ensure our responses have adequately addressed the reviewers' concerns. Please let us know if there are any remaining questions or areas where further clarification is needed. We sincerely appreciate the time and effort you have dedicated to reviewing our work.
>
> Kind regards,
>
> Authors

---

### Official Review · Reviewer_ETsw · 2024-11-06

**Soundness:** 3
**Presentation:** 3
**Contribution:** 3
**Rating:** 6
**Confidence:** 4

**Summary:**

The paper investigates an RL method for variable selection in branch-and-bound. It applies the general definition of MDP to B&B search so that general RL algorithms and theories can be applied to solve the problem. The MDP is solved with Q-learning. In experiments, the method is evaluated on four standard MILP benchmarks and compared against other RL methods. It achieve the best results compared to other RL methods, but still worse than IL methods.

**Strengths:**

1. The paper developed BBMDP tailored for B&B search that allows the application of a boarder range of developed RL methods.
2. The empirical results look promising in comparison to other RL methods.
3. The paper is well-written and easy to follow.

**Weaknesses:**

I don’t see major weaknesses, but have a couple of questions that could be of potential improvement to the paper:
1. Can you show the sample complexity of the methods compared to previous ones?
2. Is your methods also effective on larger instances? For instances that cannot be solved within the runtime limit, one can still evaluate the primal-dual gap/primal-dual integral to see if the method is effective or not.

**Questions:**

See weaknesses.

---

> ### Author Response · Authors · 2024-11-22
> **Response to reviewer ETsw**
>
> We thank the reviewer for their valuable time and feedback. We address their questions in order.
>
> > Can you show the sample complexity of the methods compared to previous ones?
>
> 1. We understand that the reviewer is interested in comparing the sample efficiency of the different RL agents. We provide training curves in Appendix G, referred to line 369, which show that sample efficiency is roughly equivalent between DQN-BBMDP, DQN-TreeMDP and DQN-Retro. We did not retrain the PG-tMDP agent from (Scavuzzo et al. 2022), however, we are confident that sample complexity would be worse than that of DQN agents, since REINFORCE is an on-policy learning algorithm, while DQN is off-policy. If we misunderstood the reviewer’s point, could the reviewer please reformulate their question.
>
>
> >Is your methods also effective on larger instances? For instances that cannot be solved within the runtime limit, one can still evaluate the primal-dual gap/primal-dual integral to see if the method is effective or not.
>
> 2. Thank you for raising this important issue. Since no prior RL contribution tested its agents on higher-dimensional problems, for the sake of comparison, we did not carry out tests on harder instance benchmarks at first. In fact, RL agents more than IL agents have been found to struggle to generalize to higher dimensions, which is why prior contributions only evaluated generalization on medium benchmarks. Nevertheless, we will include performance comparison between RL and IL agents on the hard benchmark shortly.

---

> ### Author Response · Authors · 2024-12-02
> **Follow up**
>
> Dear Reviewer ETsw,
>
> Thank you again for your valuable feedback. We have conducted additional experiments on medium and hard instances that we integrated to Appendix H, along with additional performance metrics. As the rebuttal period is nearing its conclusion, we wanted to follow up to ensure our responses have adequately addressed the reviewers' concerns. Please let us know if there are any remaining questions or areas where further clarification is needed. We appreciate the time and effort you have dedicated to reviewing our work.
>
> Sincerely,
>
> Authors

---

### Official Review · Reviewer_hrBM · 2024-11-07

**Soundness:** 3
**Presentation:** 1
**Contribution:** 2
**Rating:** 3
**Confidence:** 3

**Summary:**

This paper studies learning efficient branching strategies by reinforcement, and introduces B&B MDPs, a principled vanilla MDP formulation for variable selection, allowing to leverage a broad range of RL algorithms for the purpose of learning optimal B&B heuristics. The proposed method defines a Bellman optimality operator to unlock the full potential of approximate dynamic programming algorithms. On easy instances, DQNBBMDP consistently obtains best performance among RL agents

**Strengths:**

This paper overperforms the previous RL agents while narrowing the gap with the IL approach.

**Weaknesses:**

1. The primary innovations of BBMDP are unclear.
2. The experiments are conducted on a relatively small scale. The medium instances require only 1 minute to solve.
3. BBMDP's performance is not significantly superior to other RL methods. It is even inferior on the medium dataset.

**Questions:**

1. Could you clearly differentiate between BBMDP and TreeMDP?
2. Regarding the statement: “Yet, if the performance of IL heuristics are capped by that of the suboptimal branching experts they learn from, the performance of RL branching strategies are, in theory, only bounded by the maximum score achievable.” What does "suboptimal branching experts" refer to? Is it the strong branching rule?
3. On page 2, line 91, there's a mention of “Since ρ necessarily defines a total order on nodes.” Given that node selection and variable selection influence each other, what does it mean to have this fixed order?
4. Could you conduct tests on larger datasets, similar to those used by Gasse et al.?
5. Would it be possible to provide additional metrics, such as wins and P-D convergence plots?

---

> ### Author Response · Authors · 2024-11-22
> **Response to reviewer hrBM**
>
> We thank the reviewer for their valuable time and feedback. We have made modifications to the paper that we hope improve clarity and presentation. We address each concern in order:
>
> > The primary innovations of BBMDP are unclear.
>
> 1. We thank the reviewer for sharing their concerns on the primary innovations of BBMDP. We added a general comment that we hope helps clarify our contribution.
>
> >The experiments are conducted on a relatively small scale. The medium instances require only 1 minute to solve.
>
> 2. We thank the reviewer for raising this important issue. Since no prior RL contribution tested its agents on higher-dimensional problems, for the sake of comparison, we did not carry out tests on harder benchmarks at first. In fact, RL agents more than IL agents have been found to struggle to generalize to higher-dimensional instances, which is why prior RL contributions only evaluated generalization on medium benchmarks. Nevertheless, we will include performance comparison between RL and IL agents on hard benchmarks similar to the one in (Gasse et al. 2019).
>
> > BBMDP's performance is not significantly superior to other RL methods. It is even inferior on the medium dataset.
>
> 3. We thank the reviewer for raising their concern. In fact, we believe the performance gains presented in Table 1 are substantial. As highlighted in the introduction, MILP solvers are highly optimized software, and achieving a 10–15% reduction in solving time is usually recognized as a significant achievement within the combinatorial optimization community. Additionally, due to the relatively small size of our training instances, surpassing existing solvers becomes even more challenging, as these solvers excel at solving lower-dimensional instances. In order to make our results more meaningful, we include two additional columns in Table 1 p.9, showing the aggregate average performance obtained by each agent across the four MILP benchmarks, normalized by the score of DQN-BBMDP. It appears that the performance gap between IL and DQN-BBMDP is clearly narrower than the gap between DQN-BBMDP and any other RL agent. This is particularly true when evaluating generalization performance on the medium dataset, as discussed line 436-438 in the latest version, which underpins the computational benefit of adopting BBDMP over TreeMDP.
>
> We also address the questions explicitly raised by the reviewer.
>
> >Could you clearly differentiate between BBMDP and TreeMDP?
> 1. The main difference between BBMDP and TreeMDP, is that BBMDP is a MDP, while TreeMDP is not. We included clearer comparisons between BBMDP and TreeMDP in section 3.1 line 216-219, as well as in the introduction, line 50-55, in order to make it more explicit for the reader. In fact, the BBMDP framework allows to leverage a broader range of RL algorithms in order to learn efficient branching strategies, while preserving the convergence properties brought by TreeMDP’s use of DFS node selection policy.  In particular, this is achieved by defining the entire B&B tree as the current state, instead of merely the current B&B node as in TreeMDP.
>
>  >What does "suboptimal branching experts" refer to? Is it the strong branching rule?
>
> 2. Yes, this is strong branching. We added a paragraph on imitation learning in the Related Work section 2.3, line 152-159, in order to make this more explicit for the reader.
>
> >Given that node selection and variable selection influence each other, what does it mean to have this fixed order?
>
> 3. We thank the reviewer for pointing out this issue. Ideally, variable and node selection policies should be optimized jointly in Equation (1) in order to fully optimize the performance of B&B over a distribution of instances. Following findings of (Etheve 2021), who found node selection optimization to be negligible against variable selection optimization, we focus on improving variable selection. However, we argue that our BBMDP framework is a necessary step towards the joint optimization of node and variable selection strategies. In fact, node and variable selection strategies can be optimized jointly within the BBMDP framework.
> In fact, this is simply achieved by redefining action selection as the process of selecting both an open node and an integer variable to branch on.  Such a reformulation is not possible within the TreeMDP framework, as states are defined as the current B&B nodes, which, by definition, are determined by the node selection policy, and can hence not be governed by the action selection policy.
>
> >Could you conduct tests on larger datasets, similar to those used by Gasse et al.?
>
> 4. At the reviewer’s request, we will include tests on larger datasets, similar to the ones in Gasse et al. (2019).
>
> >Would it be possible to provide additional metrics, such as wins and P-D convergence plots?
>
> 5.  At the reviewer’s request, we will provide additional performance metrics in the Appendix.

---

> ### Author Response · Authors · 2024-12-02
> **Follow up**
>
> Dear Reviewer hrBM,
>
> Thank you again for your valuable feedback. We have conducted additional experiments on medium and hard instances that we integrated to Appendix H, along with additional performance metrics. We also provided thorough discussions to address your concerns. As the rebuttal period is nearing its conclusion, we wanted to follow up to ensure our responses have adequately addressed the reviewers' concerns. Please let us know if there are any remaining questions or areas where further clarification is needed. We appreciate the time and effort you have dedicated to reviewing our work.
>
> Sincerely,
>
> Authors

---

### Author Response · Authors · 2024-11-22
**BBMDP primary innovations**

Three reviewers have expressed concerns about the primary innovations of BBMDP, particularly in relation to the TreeMDP framework. These general concerns are addressed in this section. To put it shortly, the main difference between BBMDP and TreeMDP, is that BBMDP is a MDP, while TreeMDP is not. In fact, BBMDP, preserves the training convergence properties brought by the TreeMDP framework, along with fundamental MDP properties. We believe that leveraging the BBMDP framework is essential for advancing learning-based variable selection strategies in the future, as it enables the integration of state-of-the-art MCTS model-based RL algorithms that have proven capable of surpassing human knowledge in combinatorial tasks.

When training neural networks to learn efficient branching strategies, the choice of state representation function is key. In their seminal work, Gasse et al. (2019) introduced bipartite graph representations for encoding MILPs, demonstrating their effectiveness in learning to imitate strong branching decisions at each node of the branch-and-bound process. In TreeMDPs, Scavuzzo et al. (2022) define states as B&B nodes, allowing to reuse the bipartite graph representation and graph convolutional architecture introduced by Gasse et al. (2019) to learn efficient branching strategies by reinforcement. However, TreeMDPs are not MDPs, as they do not define a Markov process on the state random variable: a TreeMDP transition yields two future states, and is hence not a stochastic process on the state variables. In our work, we define BBMDP, a principled MDP formulation for variable selection in B&B. In particular, we define state as the entire B&B tree, not just the current node. Consequently, we cannot use MILP bipartite graphs to represent MDP states, nor use graph convolutional neural networks to learn policies or value functions directly. A potential solution is to represent B&B trees as hypergraphs, where each B&B node is encoded by its corresponding MILP bipartite graph. However, this approach faces two significant challenges: designing an efficient function approximator for such a complex data structure, and mitigating the redundancy in this representation, which leads to significant computational overhead. This is one of the reasons why (Etheve et al. 2020) and (Scavuzzo et al. 2022) considered the standard MDP framework impractical for learning branching strategies. In Section 3.2, we propose a principled solution to this issue by demonstrating that, in a DFS context, the V-value of a BBMDP state can be decomposed as the sum of the W-values of all its open nodes. Additionally, we show that these W-value functions can be learned by traditional approximate value iteration algorithms. As demonstrated in Section 3.3, this allows to train GCNNs via Q-learning to approximate proxy value functions, using bipartite graphs to represent B&B nodes, and ultimately to derive greedy policies for BBMDPs. In Section 3.4, we highlight the advantages of BBMDP over TreeMDP by presenting a concrete example in which TreeMDP results in inconsistent learning schemes. Finally, our computational experiments presented in Section 4 further supports the adoption of BBMDP over TreeMDP, as the DQN-BBMDP agent clearly outperforms TreeMDP agents on both training and generalization instances of the Ecole benchmark.

---

### Author Response · Authors · 2024-11-27
**Furter computational results**

Three reviewers have inquired about the generalization capacity of our agent on more challenging instance benchmarks. In response to the reviewers' request, in Appendix H, we include further computational results on hard instance benchmarks as defined in Gasse et al. (2019). We observe that while the IL expert demonstrates reasonable generalization capability, achieving performance comparable to SCIP on hard instance benchmarks, all RL baselines perform poorly, with none managing to outperform the random agent across the four benchmarks, as shown in Table 5 p.21. This underscores the limited generalization capacity of current model-free RL agents to higher-dimensional instances, and highlights the need to adapt model-based RL approaches to the B&B setting, by leveraging the BBMDP framework.

In response to the reviewers' request, we have also updated the computational results for the easy and medium instance benchmarks presented in Table 1, now including performance scores obtained across 100 instances and 5 seeds. These computational results are consistent with the ones presented in prior versions. Per-benchmark standard deviations can be found in Table 7 in Appendix H.

Finally, in order to better differentiate between baselines across easy, medium and hard instance benchmarks, we include additional performance metrics such as wins and average baseline rank, as shown in Table 6. The performance ranking derived from these supplementary metrics is consistent with the one discussed in Section 4.3, based on the computational results presented in Table 1.

---

### Meta-Review · Area_Chair_4auU · 2024-12-24

**Metareview:**

The paper describes a new RL technique to obtain a good branching heuristic for B&B applied to ILP problems.  It claims to improve the state of the art of RL techniques for branching.  The strengths of the paper include a challenging problem and some interesting ideas.  The main weakness of the paper is the weak performance of the approach that is worse than some other RL techniques on hard instances and worse than imitation learning in general.  The rebuttal and the revised paper clarified many things and improved substantially the experiments.  However, given that the main claim of the paper is that the proposed technique improves on previous RL techniques, but the results on hard instances show the opposite, this work is not ready for publication.  Just to be clear, this is not just a question of results, but there needs to be a clear justification for why the proposed technique should improve with respect to RL alternatives.  The paper suggests that the proposed technique is based on a proper MDP formulation in contrast to previous formulations, but does this does not seem to help empirically.  Similarly, the paper justifies the investigation of RL techniques on the basis that they are not limited to a suboptimal expert, but the imitation learning techniques still dominate.  This means that some other factor is at play.  The authors are encouraged to understand what are the main factors of success that help imitation learning and other RL techniques outperform their technique in hard instances.

**Additional Comments On Reviewer Discussion:**

There was no additional discussion among the reviewers.

---

### Decision · Program_Chairs · 2025-01-22

Reject